# Random Noise Defense Against Query-Based Black-Box Attacks

**Zeyu Qin**[1]    **Yanbo Fan**[2]    **Hongyuan Zha**[1]    **Baoyuan Wu**[1][†]
[1]School of Data Science, Shenzhen Research Institute of Big Data,
The Chinese University of Hong Kong, Shenzhen
[2]Tencent AI Lab
`zeyuqin@link.cuhk.edu.cn, fanyanbo0124@gmail.com,`
`zhahy@cuhk.edu.cn, wubaoyuan@cuhk.edu.cn`

## Abstract

The query-based black-box attacks have raised serious threats to machine learning models in many real applications. In this work, we study a lightweight defense method, dubbed *Random Noise Defense* (RND), which adds proper Gaussian noise to each query. We conduct the theoretical analysis about the effectiveness of RND against query-based black-box attacks and the corresponding adaptive attacks. Our theoretical results reveal that the defense performance of RND is determined by the magnitude ratio between the noise induced by RND and the noise added by the attackers for gradient estimation or local search. The large magnitude ratio leads to the stronger defense performance of RND, and it's also critical for mitigating adaptive attacks. Based on our analysis, we further propose to combine RND with a plausible Gaussian augmentation Fine-tuning (RND-GF). It enables RND to add larger noise to each query while maintaining the clean accuracy to obtain a better trade-off between clean accuracy and defense performance. Additionally, RND can be flexibly combined with the existing defense methods to further boost the adversarial robustness, such as adversarial training (AT). Extensive experiments on CIFAR-10 and ImageNet verify our theoretical findings and the effectiveness of RND and RND-GF.

## 1    Introduction

Deep neural networks (DNNs) have been successfully applied in many safety-critical tasks, such as autonomous driving, face recognition and verification, *etc*. However, it has been shown that DNN models are vulnerable to adversarial examples [18, 21, 26, 29, 48], which are indistinguishable from natural examples but make a model produce erroneous predictions. For real-world applications, the DNN model as well as the training dataset, are often hidden from users. Instead, only the model feedback for each query (*e.g.*, labels or confidence scores) is accessible. In this case, the product providers mainly face severe threats from query-based black-box attacks, which don't require any knowledge about the attacked models.

In this work, we focus on efficient defense techniques against query-based black-box attacks, of which the main challenges are **1)** the defender should not significantly influence the model's feedback to normal queries, but it is difficult to know whether a query is normal or malicious; **2)** the defender has no information about what kinds of black-box attack strategies adopted by the attacker. Considerable efforts have been devoted to improving the adversarial robustness of DNNs [13, 34, 49, 50]. Among them, adversarial training (AT) is considered as the most effective defense techniques [3, 49]. However, the improved robustness from AT is often accompanied by significant degradation of the

---

[†]Corresponding author.

35th Conference on Neural Information Processing Systems (NeurIPS 2021).

clean accuracy. Besides, the training cost of AT is much higher than that of standard training, and then it also often suffers from poor generalization to new samples or adversarial attacks [20, 39, 45, 46]. Thus, we argue that AT-based defense is not a very suitable choice for black-box defense. In contrast, we expect that a good defense technique should satisfy the following requirements: *well keeping clean accuracy, lightweight, and plug-and-play*.

To this end, we study a lightweight defense strategy, dubbed *Random Noise Defense* (RND) against query-based black-box attacks. For query-based attacks [1, 2, 7, 9, 11, 19, 23, 26, 27, 32, 35], the core is to find an attack direction by gradient estimation or random search based on the exact feedback of consecutive queries, which leads to a decrease of the designed objective. RND is realized by adding random noise to each query at the inference time. Therefore, the returned feedback with randomness results in poor gradient estimation or random search and slows down the attack process.

To better understand the effectiveness of RND, we provide a theoretical analysis of the defense performance of RND against query-based attacks and adaptive attacks. Our theoretical results reveal that the defense performance of RND is determined by the magnitude ratio between the noise induced by RND and the noise added by the attackers for gradient estimation or random search. The attack efficiency is significantly affected by a large magnitude ratio. The attackers need more queries to find the adversarial examples or fail to find a successful attack under limited query settings. That is, the larger ratio leads to the better defense performance of RND. Apart from standard attacks, the adaptive attack (EOT) [3] has been considered as an effective strategy to mitigate the random effect induced by the defenders. We also conduct theoretical analysis about the defense effect of RND against EOT attacks, and demonstrate that the magnitude ratio is also crucial to mitigate adaptive attacks and the adaptive attacks have the limited impact of evading the RND. On the other hand, large random noises to each query may lead to degradation of clean accuracy. To achieve a better trade-off between the defense effect and the clean accuracy while maintaining training time efficiency, we further propose combining RND with a lightweight Gaussian augmentation Fine-tuning (RND-GF). RND-GF enables us to adopt larger noise in inference time to disturb the query process better.

We conduct extensive experiments on CIFAR-10 and ImageNet. The experimental results verify our theoretical results and demonstrate the effectiveness of RND-GF. It is worth noting that RND is plug-and-play and easy to combine with existing defense methods such as AT to boost the defense performance further. To verify these, we evaluate the performance of RND combined with AT [22] and find that the RND can improve the robust accuracy of AT by up to $23.1\%$ against the SOTA black-box attack Square attack [2] with maintaining clean accuracy.

The main contributions of this work are four-fold.

- We study a lightweight random noise defense (RND) against black-box attacks theoretically and empirically. Our theoretical results reveal that the effectiveness of RND is determined by the magnitude ratio between the noise induced by RND and the noise added by the attackers for gradient estimation and random search.

- We theoretically analyze the performance of RND against the adaptive attack (EOT) and demonstrate that EOT has the limited effect of evading the RND.

- Leveraging our theoretical analysis, we further propose an efficient and stronger defense strategy RND-GF by combining the Gaussian augmentation Fine-tuning and RND towards a better trade-off between clean and adversarial performance.

- Extensive experiments verify our theoretical analysis and show the effectiveness of our defense methods against several state-of-the-art query-based attacks.

## 2 Related Work

**Query-based Methods.**   Here we mainly review the query-based black-box attack methods, which can be categorized into two classes, including ***gradient estimation*** and ***search-based*** methods. Gradient estimation methods are based on zero-order (ZO) optimization algorithms. The attacker utilizes a direct search strategy to find the search direction in search-based methods instead of explicitly estimating gradient. Furthermore, for our theoretical analysis, we focus on ***score-based*** queries. The continuous score (*e.g.*, the posterior probability or the logit) for each query is returned, in contrast to the decision-based queries which return hard labels. Specifically, [26] proposed the first limited query-based attack method by utilizing the Natural Evolutionary Strategies (NES) to estimate the

gradient. [32] proposed the ZOsignSGD algorithm, which is similar to NES. Based on Bandit Optimization, [27] proposed to combine the time and data-dependent gradient prior with gradient estimation, which dramatically reduced the number of queries. For the $\ell_2$ norm constrain, SimBA [23] randomly sampled a perturbation from orthonormal basis. SignHunter [1] focuses on estimating the sign of gradient and flipped the sign of perturbation to improve query efficiency. Square attack [2] is the state-of-art query-based attack method that selects localized square-shaped updates at random positions of images.

**Black-Box Defense.** Compared with the defense for white-box attacks, the defense specially designed for black-box attacks has not been well studied. Two recent works [8, 31] proposed to detect malicious queries based on the comparison with the history queries since the malicious queries are more similar with each other compared with normal queries. AdvMind [37] proposed to infer the intent of the adversary, and it also needed to store the history queries. However, suppose the attacker adopted the strategy of long-interval malicious queries. In that case, the defender has to store a long history, with a very high cost of storage and comparison. [42] proposed the first black-box certified defense method, dubbed denoised smoothing. [5] also showed that adding random noise can defend against query-based attacks through experimental evaluations, without theoretical analysis of the defense effect. There are also a few randomization-based defenses. R&P [51] proposed a random input transform-based method. RSE [33] added large Gaussian noise to both the input and activation of each layer. PNI [25] combined adversarial training with adding Gaussian noise to the input or weight of each layer. However, these methods significantly sacrificed the accuracy of benign examples and also have a huge training cost. PixelDP [30] and random smoothing [13, 41] proposed to train classifiers with large Gaussian noise to get certified robustness under the $\ell_2$ norm. However, they require to obtain a majority prediction of this query. Obviously, that places a huge burden on model inference and even helps the attacker get the more accurate gradient estimation. The too large Gaussian noise used in those methods also sacrifices clean accuracy. In contrast, RND maintains a good clean accuracy and only perturbs each query once, without any extra burden. [40] showed that the model with Gaussian augmentation training achieves the state-of-art defense against common corruptions, while the defense to black-box attacks is not evaluated.

## 3 Preliminaries

### 3.1 Score-based Black-Box Attack

In this work, we mainly focus on the score-based attacks. The analysis and evaluation of decision-based attacks are shown in **supplementary materials**. We denote the attacked model as $\mathcal{M} : \mathcal{X} \to \mathcal{Y}$ with $\mathcal{X}$ being the input space and $\mathcal{Y}$ being the output space. Given a benign data $(\boldsymbol{x}, y) \in (\mathcal{X}, \mathcal{Y})$, the goal of adversarial attack is to generate an adversarial example $\boldsymbol{x}_{adv}$ that is similar with $\boldsymbol{x}$, but to enforce prediction of $\mathcal{M}$ to be different with the true label $y$ (*i.e.*, untargeted attack) or to be a target label $h$ (*i.e.*, targeted attack). The optimization of untargeted attack can be formulated as follows,

$$\min_{\boldsymbol{x}_{adv} \in \mathcal{N}_R(\boldsymbol{x})} f(\boldsymbol{x}_{adv}) = \min_{\boldsymbol{x}_{adv} \in \mathcal{N}_R(\boldsymbol{x})} (\mathcal{M}_y(\boldsymbol{x}_{adv}) - \max_{j \neq y} \mathcal{M}_j(\boldsymbol{x}_{adv})). \tag{1}$$

For targeted attack, the objective function is $\max_{j \neq h} \mathcal{M}_j(\boldsymbol{x}_{adv}) - \mathcal{M}_h(\boldsymbol{x}_{adv})$. $\mathcal{M}_j(\boldsymbol{x}_{adv})$ denotes the logit or softmax output *w.r.t.* class $j$. $\mathcal{N}_R(\boldsymbol{x}) = \{\boldsymbol{x}' | \|\boldsymbol{x}' - \boldsymbol{x}\|_p \leq R\}$ indicates a $\ell_p$ ball around $\boldsymbol{x}$ ($p$ is often specified as 2 or $\infty$), with $R > 0$. In attack evaluation, as long as $\mathcal{L}(\boldsymbol{x}_{adv})$ is less than 0, attackers consider the attack to be successful.

The score-based attack algorithms commonly adopt the projected gradient descent algorithms,

$$\boldsymbol{x}_{t+1} = \text{Proj}_{\mathcal{N}_R(\boldsymbol{x})}(\boldsymbol{x}_t - \eta_t g(\boldsymbol{x}_t)). \tag{2}$$

For white-box attacks, $g(\boldsymbol{x}_t)$ is the gradient of $f(\boldsymbol{x})$ *w.r.t.* $\boldsymbol{x}_t$. However, for black-box attacks, the gradient $g(\boldsymbol{x}_t)$ cannot be directly obtained. So the attackers utilize the gradient estimation or random search to conduct the update $g(\boldsymbol{x}_t)$.

### 3.2 Zero-Order Optimization for Black-Box Attack

Zero-order optimization (ZO) has become the mainstream of black-box attacks, dubbed ZO attacks. The general idea of ZO attack methods is to estimate the gradient according to the objective values returned by querying. The gradient estimator widely used in score-based and decision-based attacks

[7, 10, 12, 12, 16, 26, 27, 32, 36] is

$$g_\mu(\boldsymbol{x}) = \frac{f(\boldsymbol{x} + \mu\boldsymbol{u}) - f(\boldsymbol{x})}{\mu}\boldsymbol{u}, \tag{3}$$

where $\boldsymbol{u} \sim \mathcal{N}(\mathbf{0}, \mathbf{I})$, and $\mu \geq 0$. Based on this gradient estimator, the update of attack becomes

$$\boldsymbol{x}_{t+1} = \mathrm{Proj}_{\mathcal{N}_R(\boldsymbol{x})}(\boldsymbol{x}_t - \eta_t g_\mu(\boldsymbol{x}_t)). \tag{4}$$

## 3.3 Random Search for Black-Box Attack

The search-based attacks also depend on the value of $f(\boldsymbol{x} + \mu\boldsymbol{u}) - f(\boldsymbol{x})$ to find attack directions. Here, $\boldsymbol{u}$ is a searching direction sampled from some pre-defined distributions, such as gaussian noise in [12], orthogonal basis in [23] and squared perturbations in [2]. The $\mu$ is the size of perturbation in each search step such as the size of square in Square attack [2]. The work of [1, 35] adopt the fixed size schedule. If $f(\boldsymbol{x} + \mu\boldsymbol{u}) - f(\boldsymbol{x}) < 0$, the attackers consider $\boldsymbol{u}$ as a valuable attack direction. So, the direction searching becomes

$$s(\boldsymbol{x}) = \mathbb{I}(h(\boldsymbol{x}) < 0) \cdot \mu\boldsymbol{u} \text{ where } h(\boldsymbol{x}) = f(\boldsymbol{x} + \mu\boldsymbol{u}) - f(\boldsymbol{x}), \tag{5}$$

where, $\mathbb{I}$ is the indicator function. And the updating of search-based attacks becomes

$$\boldsymbol{x}_{t+1} = \mathrm{Proj}_{\mathcal{N}_R(\boldsymbol{x})}(\boldsymbol{x}_t + s(\boldsymbol{x}_t)). \tag{6}$$

# 4 RND: Random Noise Defense Against Query-based Attack Methods

## 4.1 Random Noise Defense

In the gradient estimator (*e.g.*, Eq.(3)) and searching direction (*e.g.*, Eq.(5)), the attacker adds one small random perturbation $\mu\boldsymbol{u}$ to get the objective difference between two queries. Thus, if the defender can further disturb this random perturbation, the gradient estimation or searching direction is expected to be misled to decrease the attack efficiency. However, it's also hard for defenders to identify whether a query is normal or malicious. Inspired by those observations, we study a lightweight defense method, dubbed Random Noise Defense (RND), that simply adds random noise to each query. For RND, the feedback for one query $\boldsymbol{x}$ is $\mathcal{M}(\boldsymbol{x} + \nu\boldsymbol{v})$, with $\boldsymbol{v} \sim \mathcal{N}(\mathbf{0}, \mathbf{I})$ is standard Gaussian noise added by the defender, and the factor $\nu > 0$ controls the magnitude of random noise. Considering the task of defending query-based black-box attacks, RND should satisfy: **1)** the prediction of each query will not be changed significantly; **2)** the estimated gradient or direction searching should be perturbed as large as possible towards better defense performance. The gradient estimator and searching direction under RND become

$$g_{\mu,\nu}(\boldsymbol{x}) = \frac{f(\boldsymbol{x} + \mu\boldsymbol{u} + \nu\boldsymbol{v}_1) - f(\boldsymbol{x} + \nu\boldsymbol{v}_2)}{\mu}\boldsymbol{u}, \tag{7}$$

$$s_\nu(\boldsymbol{x}) = \mathbb{I}(h_\nu(\boldsymbol{x}) < 0) \cdot \mu\boldsymbol{u} \text{ where } h_\nu(\boldsymbol{x}) = f(\boldsymbol{x} + \mu\boldsymbol{u} + \nu\boldsymbol{v}_1) - f(\boldsymbol{x} + \nu\boldsymbol{v}_2), \tag{8}$$

where $\boldsymbol{v}_1, \boldsymbol{v}_2 \sim \mathcal{N}(\mathbf{0}, \mathbf{I})$ are both standard Gaussian noise generated by the defender.

To satisfy the first requirement, one cannot add too large noise to each query. However, to meet the second requirement, $\nu$ should be large enough to change the gradient estimation or searching direction. We need to choose a proper $\nu$ to achieve a good trade-off between these two requirements. In the following, we provide the theoretical analysis of RND, which can shed light on the setting of $\nu$.

## 4.2 Theoretical Analysis of Random Noise Defense Against ZO Attacks

In this section, we will present the theoretical analysis of the effect of RND against ZO attacks. Specifically, we study the convergence property of ZO attack in Eq.(4) with $g_{\mu,\nu}(\boldsymbol{x})$ in Eq.(7) being the gradient estimator. Throughout our analysis, the measure of adversarial perturbation is specified as $\ell_2$ norm, corresponding to $\mathcal{N}_R(\boldsymbol{x}) = \{\boldsymbol{x}'|\|\boldsymbol{x}' - \boldsymbol{x}\|_2 \leq R\}$. To facilitate subsequent analyses, we first introduce some assumptions, notations, and definitions.

**Assumption 1.** $f(\boldsymbol{x})$ *is Lipschitz-continuous,* i.e., $|f(\boldsymbol{y}) - f(\boldsymbol{x})| \leq L_0(f)\|\boldsymbol{y} - \boldsymbol{x}\|$.

**Assumption 2.** $f(\boldsymbol{x})$ *is continuous and differentiable, and* $\nabla f(\boldsymbol{x})$ *is Lipschitz-continuous,* i.e., $\|\nabla f(\boldsymbol{y}) - \nabla f(\boldsymbol{x})\| \leq L_1(f)\|\boldsymbol{y} - \boldsymbol{x}\|$.

**Notations.** We denote the sequence of standard Gaussian noises added by the attacker as $\mathcal{U}_t = \{\boldsymbol{u}_0, \boldsymbol{u}_1, \ldots, \boldsymbol{u}_t\}$, with $t$ being the iteration index in the update Eq.(4). The sequence of standard Gaussian noises added by the defender is denoted as $\mathcal{V}_t = \{\boldsymbol{v}_{01}, \boldsymbol{v}_{02}, \ldots, \boldsymbol{v}_{t1}, \boldsymbol{v}_{t2}\}$. The generated sequential solutions are denoted as $\{\boldsymbol{x}_0, \boldsymbol{x}_1, \ldots, \boldsymbol{x}_Q\}$, and the benign example $\boldsymbol{x}$ is used as the initial solution, *i.e.*, $\boldsymbol{x}_0 = \boldsymbol{x}$. $d = |\mathcal{X}|$ denotes the input dimension.

**Definition 1.** *The Gaussian-Smoothing function corresponding to $f(\boldsymbol{x})$ with $\nu > 0, \boldsymbol{v} \sim \mathcal{N}(\mathbf{0}, \mathbf{I})$ is*

$$f_\nu(\boldsymbol{x}) = \frac{1}{(2\pi)^{d/2}} \int f(\boldsymbol{x} + \nu\boldsymbol{v}) \cdot \mathrm{e}^{-\frac{1}{2}\|\boldsymbol{v}\|_2^2} \, \mathrm{d}\boldsymbol{v}. \tag{9}$$

Due to the noise inserted by the defender, $f_\nu$ becomes the objective function for the attacker. We also define the minimum of $f_\nu(\boldsymbol{x})$, $f_\nu^* = f_\nu(\boldsymbol{x}^*) = \min_{\boldsymbol{x} \in \mathcal{N}_R(\boldsymbol{x}_0)} f_\nu(\boldsymbol{x})$.

### 4.2.1 Theoretical Analysis under General Non-Convex Case

Here, we report theoretical analysis of RND under general non-convex case satisfying Assumption 1.

**Theorem 1.** *Under Assumption 1, for any $Q \geq 0$, consider a sequence $\{\boldsymbol{x}_t\}_{t=0}^Q$ generated according to the descent update Eq.(4) using the gradient estimator $g_{\mu,\nu}(\boldsymbol{x})$ in Eq.(7), with the constant stepsize,*
$\eta = \left[\frac{R\epsilon}{\gamma(\alpha)^2 d^3 L_0^3(f)(Q+1)}\right]^{1/2}$ *Then,*

$$\frac{1}{Q+1}\sum_{t=0}^Q \mathbb{E}_{\mathcal{U}_t, \mathcal{V}_t}(\|\nabla f_{\mu,\nu}(\boldsymbol{x}_t)\|^2) \leq \frac{2L_0(f)^{\frac{5}{2}} R^{\frac{1}{2}} d^{\frac{3}{2}}}{(Q+1)^{\frac{1}{2}}\epsilon^{\frac{1}{2}}}\gamma(\alpha) \tag{10}$$

*where $\frac{\nu}{\mu} = \alpha$ and $\gamma(\alpha) = \alpha + \frac{\sqrt{2}}{2}$, which is always increasing function of $\alpha$.*

**Remark 1.** *Due to the non-convexity assumption, we only guarantee the convergence to a stationary point of the function $f_{\mu,\nu}(\boldsymbol{x})$, which is a smoothing approximation of $f_\nu$. To bound the gap $\epsilon$ between $f_{\mu,\nu}(\boldsymbol{x})$ and $f_\nu(\boldsymbol{x})$, i.e., $|f_{\mu,\nu}(\boldsymbol{x}) - f_\nu(\boldsymbol{x})| \leq \epsilon, \forall \boldsymbol{x} \in \mathcal{N}_R(\boldsymbol{x})$, we could choose $\mu \leq \hat{\mu} = \frac{\epsilon}{d^{1/2}L_0(f)}$ [36]. The minimum of upper bound is denoted as $\delta$, then the upper bound for the expected number of queries is $O\left(\gamma(\alpha)^2 \frac{d^3 L_0^5(f) R}{\epsilon \delta^2}\right)$.*

The convergence rate of ZO attacks is important since attackers need to find adversarial examples within limited queries effectively. Theorem 1 shows the convergence rate is positive related to the ratio $\frac{\nu}{\mu}$. The larger ratio $\frac{\nu}{\mu}$ will lead to the higher upper bound of convergence error and slower convergence rate. In consequence, the attackers need much more queries to find adversarial examples. Specifically, if $\frac{\nu}{\mu} \ll 1$, the query complexity is equivalent to the original constant term $O(\frac{d^3}{\epsilon\delta^2})$. When $\frac{\nu}{\mu} \geq 1$, the query complexity is really improved over $O(\frac{d^3}{\epsilon\delta^2})$. Therefore, the attack efficiency will be significantly reduced under the queries limited setting, leading to failed attacks or a much larger number of queries for successful attacks. Therefore, **the large ratio $\frac{\nu}{\mu}$ leads the effectiveness of RND.** In accordance with our intuition, the defenders should insert larger noise (*i.e.*, $\nu$) than that added by the attack (*i.e.*, $\mu$) to achieve the satisfied defense effect.

**Trade-off of Larger $\nu$ and Clean Accuracy:** If $f(\boldsymbol{x})$ is Lipschitz-continuous, then $|f_\nu(\boldsymbol{x}) - f(\boldsymbol{x})| \leq \nu L_0(f) d^{1/2}$ [36]. The larger $\nu$ is, the larger the gap between $f_\nu(\boldsymbol{x})$ and $f(\boldsymbol{x})$. So the clean accuracy of the model with adding larger noise will also decrease. This is also shown in Figure 1 (d), which forms a trade-off between defense performance of RND and clean accuracy.

**Larger Noise Size $\mu$ Adopted by Attackers:** The attacker may be aware of the defense mechanism, increasing the adopted noise size $\mu$. Our experiment results also verify this point. As shown in Figure 2 (a), for NES attack, the attack failure rate is almost 0, when $\nu = \mu = 0.01$. This shows adding small noise can't always guarantee an effective defense. Previous work [5, 15] don't consider this and overestimate the effect of small noise. However, increasing the noise size $\mu$ will also lead to less accurate gradient estimation and random search in Eq.(3) and Eq.(5), leading to a significant decrease in attack performance consequentially. Taking the NES [26] and Bandit attack [27] on ImageNet as examples, as shown in Figure 3, when $\mu$ increase from 0.01 to 0.05, the $\ell_\infty$ attack failure rates increase from $5.1\%, 5.0\%$ to $22.8\%, 25.7\%$ respectively. For Square attack, [2], when $\mu$ increases from 0.3 to 0.5, the average number of queries has doubled. To guarantee a successful

attack, attackers cannot always increase $\mu$. Our experimental results show that the attack performance decreases significantly when the $\mu$ of ZO attacks and Square attack is larger than 0.01 and 0.3. Based on the above analysis, we propose the stronger defense method in Section 4.4. It enables us to add larger noise ($\nu > \mu$) and still maintains a good clean accuracy.

### 4.2.2 Theoretical Analysis of Random Noise Defense Against Adaptive Attacks

As suggested in recent studies of robust defense [3, 6, 49], the defender should take a robust evaluation against the corresponding adaptive attack. The attacker is aware of the defense mechanism. Here we study the defense effect of RND against adaptive attacks. Since the idea of RND is to insert random noise to each query, an adaptive attacker could utilize Expectation Over Transformation (EOT) [4] to obtain a more accurate estimation, *i.e.*, querying one sample multiple times to get the average value. Then, the gradient estimator used in ZO attacks Eq.(4) becomes

$$\tilde{g}_{\mu,\nu}(\boldsymbol{x}) = \frac{1}{M} \sum_{j=1}^{M} \frac{f(\boldsymbol{x} + \mu\boldsymbol{u} + \nu\boldsymbol{v}_{j1}) - f(\boldsymbol{x} + \nu\boldsymbol{v}_{j2})}{\mu} \boldsymbol{\mu}, \tag{11}$$

where $\boldsymbol{v}_{j1}, \boldsymbol{v}_{j2} \sim \mathcal{N}(\boldsymbol{0}, \mathbf{I})$, with $j = 1, \ldots, M$. Note that here the definition of the sequential standard Gaussian noises added by the defender (see Section 4.2.1) should be updated to $\mathcal{V}_t = \{\boldsymbol{v}_{01}, \ldots, \boldsymbol{v}_{0M}, \ldots, \boldsymbol{v}_{t1}, \ldots, \boldsymbol{v}_{tM}\}$. $\boldsymbol{v}_{ij} \in \mathcal{V}_t$ contains $\boldsymbol{v}_{ij1}$ and $\boldsymbol{v}_{ij2}$. The convergence analysis of ZO attack with Eq.(11) against RND is presented in Theorem 2.

**Theorem 2.** *Under Assumption 1 and 2, for any $Q \geq 0$, consider a sequence $\{\boldsymbol{x}_t\}_{t=0}^{Q}$ generated according to the descent update Eq.(4) using the gradient estimator $\tilde{g}_{\mu,\nu}(\boldsymbol{x})$ Eq.(11), we have*

$$\frac{1}{S_Q} \sum_{t=0}^{Q} \eta_t \mathbb{E}_{\mathcal{U}_t, \mathcal{V}_t} (\|\nabla f_{\mu,\nu}(\boldsymbol{x}_t)\|^2) \leq \frac{L_0(f)R}{S_Q} + \frac{1}{S_Q} \sum_{t=0}^{Q} \eta_t^2 (L_0(f)^2 L_1(f) d^2 (\frac{1}{2} + \frac{2\nu^2}{\mu^2 M})$$
$$+ \frac{\nu^2 L_0(f) L_1(f)^2}{\mu} d^{\frac{5}{2}} + \frac{\nu^4 L_1(f)^3 (M+1)}{2\mu^2 M} d^3) \tag{12}$$

**The larger $M$ for EOT:** Theorem 2 shows that the upper bound will be reduced with larger $M$. So, EOT can mitigate the defense effect caused by the randomness of RND. However, with $M$ going to infinity, the upper bound of expected convergence error (*i.e.*, Eq. (12)) is still determined by the max term $\frac{\nu^4}{\mu^2} d^3$. **It implies that the attack improvement from EOT is limited, especially with the larger ratio $\frac{\nu}{\mu}$.** Experiments in Section 5.3 verify our analysis of EOT attack.

### 4.3 Theoretical Analysis of Random Noise Defense Against Search-based Attacks

In this section, we will show how RND affects search-based attacks. Based on the Eq.(5) and Eq.(8), by adding noise $\nu\boldsymbol{v}$, the value of $h_\nu(\boldsymbol{x})$ will be different from that of $h(\boldsymbol{x})$, and there is certain probability that $\text{Sign}(h_\nu(\boldsymbol{x}))$ be different from $\text{Sign}(h(\boldsymbol{x}))$. When the random noise $\nu\boldsymbol{v}$ causes inconsistence between $\text{Sign}(h_\nu(\boldsymbol{x}))$ and $\text{Sign}(h(\boldsymbol{x}))$, RND will mislead the attackers to select the incorrect attack directions (*i.e.*, abandoning the descent direction *w.r.t.* $f$ or selecting the ascent direction), so as to decrease attack performance. We have following theoretical analysis about this.

**Theorem 3.** *Under Assumption 1, considering the direction update Eq.(6) with Eq.(8) in search-based attacks, we have,*

$$P(Sign(h(\boldsymbol{x})) \neq Sign(h_\nu(\boldsymbol{x}))) \leq \frac{2L_0(f)\nu\sqrt{d}}{|h(\boldsymbol{x})|} \tag{13}$$

The probability $P$ is intuitively controlled by the relative values of the $h_\nu(\boldsymbol{x})$ and $h(\boldsymbol{x})$. The proof is shown in Section F of supplementary materials. Theorem 3 shows the probability of misleading attacker is positive correlated with $\frac{\nu}{|h(\boldsymbol{x})|}$. Due to the small value $\mu$ and local linearity of model, we have $|h(\boldsymbol{x})| = |f(\boldsymbol{x} + \mu\boldsymbol{u}) - f(\boldsymbol{x})| \approx L_0(f)\mu\|\boldsymbol{u}\|$. Therefore, the $|h(\boldsymbol{x})|$ is also positively correlated with the stepsize $\mu$ within the small neighborhoods. So the probability of changing the sign is positively correlated with $\frac{\nu}{\mu}$. **The larger ratio $\frac{\nu}{\mu}$ leads to the larger upper bound of the probability**. The attackers are more likely to be misleading and need much more queries to find adversarial examples. So, the defense performance of RND is better. As shown in Figure 1 (a-c), the evaluations on Square attack verify our analysis.

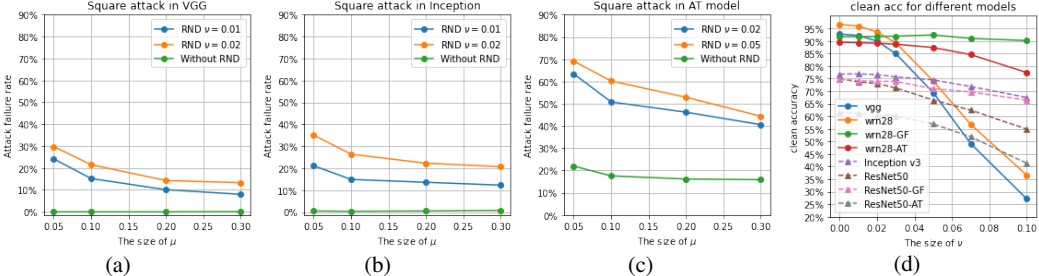

Figure 1: (a-c): Attack failure rate (%) of Square $\ell_\infty$ attacks on VGG-16(CIFAR-10), Inception v3(ImageNet) and AT model (ImageNet) under different values of $\mu$ and $\nu$, where $\mu$ is the square size in Square attacks. (d): clean accuracy for different models on CIFAR-10 and ImageNet. The circle lines and triangle lines represent models on CIFAR-10 and ImageNet respectively.

### 4.4 RND with Gaussian Augmentation Fine-tuning

The aforementioned theoretical analyses reveal that RND should choose a proper noise magnitude $\nu$ to achieve a good balance between clean accuracy and defense effectiveness. To achieve a high-quality balance, we could reduce the sensitivity of the target model to random noises. The influence of the noise on the accuracy of each query will be reduced. To satisfy the lightweight requirement of black-box defense, we propose to utilize Gaussian Augmentation Fine-tuning (GF), which fine-tunes the deployed model with Gaussian augmentation. We add random Gaussian noise to each training sample as a pre-processing step in the fine-tuning process. Consequently, the model fine-tuned with GF is expected to maintain good accuracy, even though defenders add a relatively large random noise to each query. As shown in Figure 1 (d), **GF models still maintain the high clean accuracy under larger noise**.

## 5 Experiments

### 5.1 Experimental Settings

**Datasets and Classification Models.**   We conduct experiments on two widely used benchmark datasets in adversarial machine learning: CIFAR-10 [28] and ImageNet [14]. For classification models, we use VGG-16 [43], WideResNet-16-10 and WideResNet-28-10 [54] on CIFAR-10. We conducted standard training, and their clean accuracy on the test set is 92.76% 95.10, and 96.60%, respectively. For ImageNet, we adopt the pretrained Inception v3 model [47] and ResNet-50 model [24] provided by torchvision package and the clean accuracy are 76.80% and 74.90%, respectively.

**Black-box Attack Methods and Compared Defense Methods.**   We consider several main-streamed query-based black-box attack methods, including NES [26], ZO-signSGD (ZS) [32], Bandit [27], ECO [35], SimBA [23], SignHunter [1] and Square attack [2]. Note that the NES, ZS, Bandit are gradient estimation based attack methods, and the other four are search-based methods. SimBA and ECO are only designed for $\ell_2$ and $\ell_\infty$ attack, respectively. Following [26], we evaluate all the attack methods on the whole test set of CIFAR-10 and 1,000 random sampled images from the validation set of ImageNet. We present the evaluation performance against the untargeted attack under both $\ell_\infty$ and $\ell_2$ norm settings. The perturbation budget of $\ell_\infty$ is set to 0.05 for both datasets. For $\ell_2$ attack, the perturbation budget is set to 1 and 5 on CIFAR-10 and ImageNet, respectively. The number of maximal queries is set to 10,000. **We adopt the attack failure rate and the average number of query as an evaluation metric.** The higher the attack failure rate and the larger the average number of query, the better the adversarial defense performance. We compare our methods with RSE [33] and PNI [25] on CIFAR-10. We adopt the pre-trained AT model [22] which shows better robustness than other AT models. For ImageNet, we compare pre-trained Feature Denoise (FD) model [52] and adopt pre-trained AT in Robustness Library [17]. The implementation details of those methods are given in the Section B.1 and B.2 of supplementary materials.

### 5.2 Evaluation of RND Against Query-based Black-Box Attacks

We first evaluate the defense performance of RND with various settings of $\mu$ and $\nu$ against query-based black-box attacks and verify the theoretical results in Section 4.2 and 4.3. Specifically, we set

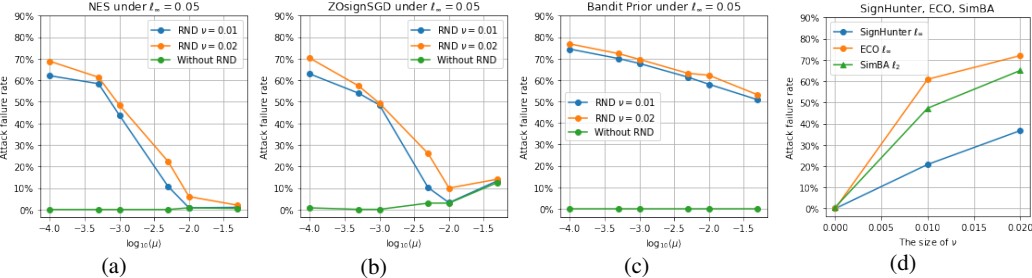

Figure 2: Attack failure rate (%) of query-based attacks on VGG-16 and CIFAR-10 under different values of $\mu$ and $\nu$. We adopt logarithm scale in subplot (a-c) for better illustration. The complete evaluation under $\ell_2$ attack is given in Section B.4 of supplementary materials.

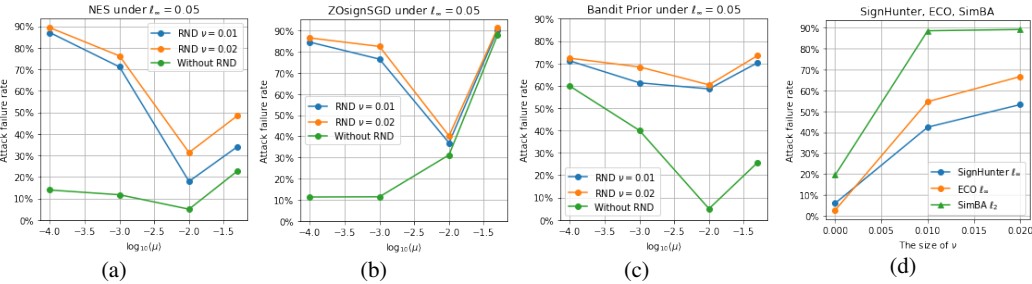

Figure 3: Attack failure rate (%) of query-based attacks on Inception v3 and ImageNet under different values of $\mu$ and $\nu$. We adopt logarithm scale in subplot (a-c) for better illustration. The complete evaluation under $\ell_2$ attack is given in Section B.4 of supplementary materials.

$\nu \in \{0.0, 0.01, 0.02\}$ and $\mu \in \{10^{-4}, 5*10^{-4}, 10^{-3}, 0.005, 0.01, 0.05\}$ for ZO attacks on CIFAR-10 and $\mu \in \{10^{-4}, 0.001, 0.01, 0.05\}$ on ImageNet. For Square attacks, we set $\mu \in \{0.05, 0.1, 0.2, 0.3\}$. Figure 1 (a-c) presents the defense performance of RND against Square attack on CIFAR-10 and ImageNet, respectively. Figure 2 (a-c) and 3 (a-c) present the defense performance of RND against NES, ZS and Bandit on CIFAR-10 and ImageNet, respectively.

The results in Figure 1 (a-c), 2 (a-c), and 3 (a-c) show that: the attack failure rate of all attack methods generally increases as the value of $\frac{\nu}{\mu}$ increases. Specifically, for a fixed value of $\nu$ (*e.g.*, 0.01), the attack failure rate increases as $\mu$ decreases. While for a certain value of $\mu$ (*e.g.*, $10^{-4}$), the attack failure rate increases as $\nu$ increases. These collaborate our theoretical analysis that **the ratio of $\frac{\nu}{\mu}$ determines the probability of changing sign and the convergence rate of ZO attacks.** The larger $\frac{\nu}{\mu}$, the higher the probability of changing the sign and the convergence error of ZO attacks, which results in poor attack performance under the query-limited settings.

We also evaluate the defense performance of RND with various values of $\nu$ against other search-based black-box attacks and the results are shown in Figure 2 (d) for CIFAR-10 and Figure 3 (d) for ImageNet. As shown in the plots, the RND can significantly boost the defense performance. The attack failure rate increases as the value of $\nu$ increases. The complete evaluations, including the WideResNet-16 on CIFAR-10, the $\ell_2$ attack, and the mean and medium number of queries of successful attacks, are reported in Section B.4 of supplementary materials.

### 5.3 Evaluation of RND Against Adaptive Attacks

We then evaluate the defense effect of RND against the adaptive attack EOT and verify our theoretical analysis in Section 4.2.2. We evaluate EOT with $M \in \{1, 5, 10\}$ on CIFAR-10 and ImageNet, and $\nu$ is set to 0.02 and $\mu$ is set to 0.001 and 0.1 for ZO attacks and Square attack, respectively. The evaluation with even larger $M$ and $\mu$ is presented in Section B.5 of supplementary materials.

We consider two settings of query budget: **1)** *adaptive query budget* that assigns query budget of 10,000 $\times M$ for different value of $M$; **2)** *fixed query budget* that adopt a fixed query budget of 10,000 for all $M$. We first evaluate the performance with adaptive query budget on NES and ZS, and their numerical results are shown in Table 1. As shown in Table 1, the attack failure rate decreases as $M$ increases on both datasets. However, the average number of the query of successful attack also significantly increases as $M$ increases, which demonstrates that **the adaptive EOT attack can**

Table 1: The evaluation of EOT with $\ell_\infty$ attack on CIFAR-10 and ImageNet under the *adaptive and fixed query setting*. **The left part is the results on CIFAR-10 and the right part is on ImageNet.** The average number of query of successful attack as well as the attack failure rate are reported. The performance of EOT with $\ell_2$ attack is reported in Section B.5 of supplementary materials.

| settings | Methods | M=1 | M=5 | M=10 | Methods | M=1 | M=5 | M=10 |
|---|---|---|---|---|---|---|---|---|
| adaptive | NES | 1448/0.484 | 4078/0.361 | 5763/0.342 | NES | 2532/0.762 | 5364/0.705 | 7582/0.691 |
| | ZS | 1489/0.493 | 3189/0.374 | 5912/0.349 | ZS | 2824/0.825 | 5735/0.761 | 7662/0.740 |
| fixed | NES | 1448/0.484 | 2528/0.452 | 3246/0.443 | NES | 2533/0.762 | 5240/0.775 | 5658/0.781 |
| | ZS | 1489/0.493 | 2765/0.448 | 3123/0.421 | ZS | 2824/0.825 | 4023/0.842 | 4652/0.861 |
| | Bandit | 436/0.696 | 276/0.582 | 314/0.543 | Bandit | 305/0.604 | 759/0.523 | 946/0.49 |
| | Square | 380/0.301 | 181/0.162 | 223/0.121 | Square | 93/0.353 | 145/0.20 | 328/0.171 |
| | SignHunter | 459/0.367 | 559/0.224 | 759/0.191 | SignHunter | 173/0.532 | 336/0.456 | 659/0.431 |
| | ECO | 904/0.720 | 1681/0.761 | 2560/0.793 | ECO | 1237/0.666 | 3065/0.678 | 3091/0.692 |
| | SimBA | 1353/0.650 | 3852/0.467 | 4103/0.396 | SimBA | 274/0.891 | 468/0.878 | 517/0.869 |

Table 2: The comparison of RND ($\nu = 0.02$), GF, RND-GF ($\nu = 0.05$), AT, RND-AT ($\nu = 0.05$), PNI, RSE, and FD on CIFAR-10 and Imagenet. **The average number of queries** of successful attack and **the attack failure rates** are reported. The best and second best attack failure rate under each attack are highlighted in bold and underlined, respectively. The evaluation under $\ell_2$ attack is shown in Section B.6 of supplementary materials.

| Datasets | Methods | Clean Acc | NES($\ell_\infty$) | ZS($\ell_\infty$) | Bandit($\ell_\infty$) | Sign($\ell_\infty$) | Square($\ell_\infty$) | SimBA($\ell_2$) | ECO($\ell_\infty$) |
|---|---|---|---|---|---|---|---|---|---|
| CIFAR-10 (WideNet-28) | Clean Model | 96.60% | 465.5/0.01 | 581.8/0.06 | 210.2/0.03 | 167.6/0.03 | 137.1/0.02 | 457.2/0.04 | 457.8/0.0 |
| | GF | 91.72% | 999.0/0.407 | 759.9/0.544 | 744.5/0.116 | 348.3/0.027 | 581.0/0.061 | 1146.8/0.395 | 883.9/0.067 |
| | RSE[33] | 84.12% | 1246.3/0.396 | 1327.8/0.422 | 281.7/0.372 | 243.7/0.221 | 413.3/0.243 | 498.3/0.337 | 578.3/0.534 |
| | PNI[25] | 87.20% | 1071.4/0.725 | 1310.7/0.823 | 324.9/0.824 | 267.0/0.708 | 295.3/0.612 | 945.0/0.857 | 2342.2/0.623 |
| | AT[22] | 89.48% | 821.6/0.807 | 614.9/0.862 | 1451.5/0.623 | 766.3/0.476 | 1135.4/0.499 | 1523.2/0.635 | 1180.4/0.484 |
| | RND | 93.60% | 842.5/0.05 | 941.8/0.143 | 273.1/0.478 | 977.2/0.226 | 762.4/0.116 | 2112.6/0.549 | 912.8/0.688 |
| | RND-GF | 92.40% | 2805.7/0.516 | 2966.3/0.730 | 1223.5/0.841 | 1017.1/0.407 | 1207.3/0.378 | 1220.2/0.863 | 687.2/0.872 |
| | RND-AT | 87.40% | 2499.2/0.842 | 2625.7/0.923 | 891.5/0.891 | 767.9/0.737 | 1170.7/0.730 | 1787.4/0.912 | 687.4/ 0.911 |
| ImageNet (ResNet-50) | Clean Model | 74.90% | 1031.9/0.0 | 2013.0/0.235 | 329.2/0.02 | 264.1/0.03 | 76.5/0.0 | 1234.5/0.281 | 347.7/0.0 |
| | GF[40] | 74.70% | 1685.5/0.03 | 1712.1/0.347 | 601.4/0.02 | 329.0/0.0 | 97.28/0.0 | 1417.4/0.112 | 362.4/0.0 |
| | FD[52] | 54.20% | 1997.2/0.679 | 1555.5/0.775 | 1579.2/0.426 | 1633.1/0.332 | 1092.4/0.242 | 2607.9/0.613 | 1501.0/0.240 |
| | AT[17] | 61.60% | 2113.4/0.724 | 1688.7/0.815 | 1091.5/0.416 | 1522.7/0.289 | 1109.0/0.159 | 2638.2/0.651 | 1440.6/0.200 |
| | RND | 73.00% | 3041.5/0.245 | 2266.2/0.330 | 390.6/0.536 | 661.0/0.314 | 81.5/0.101 | 825.3/0.612 | 2435.5/0.540 |
| | RND-GF | 71.15% | 2489.3/0.421 | 2053.5/0.563 | 495.9/0.603 | 514.0/0.348 | 1009.9/0.146 | 777.2/ 0.762 | 994.8/0.702 |
| | RND-AT | 58.15% | 2556.6/0.864 | 2596.6/0.870 | 448.0/0.810 | 724.2/0.632 | 1306.3/0.386 | 1210.5/0.953 | 631.1/0.865 |

**increase the attacking success rate with a sacrifice of query efficiency.** More importantly, we observe that **the relative performance improvements induced by EOT generally decrease as $M$ increases.** These verify our theoretical analysis in Section 4.2.2.

The evaluation under the fixed query budget is also reported in Table 1. On small-scale dataset CIFAR-10, the attack failure rate of all attacks generally decreases as $M$ increases. Yet, we also observe a similar phenomenon in that the relative performance improvements induced by EOT decrease as $M$ increases. For the large-scale dataset ImageNet, EOT can increase the attack performance with a significant sacrifice of query efficiency under most cases, except for NES and ZS. The attack performance of NES and ZS decreases as $M$ increases. Specifically, for a fixed number of queries, the larger the $M$, the smaller the number of iterations for ZO attack under limited query setting. The poor performance of NES and ZS with larger $M$ may be because the improvements induced by EOT are less than the potential degeneration caused by the decrease of iterations.

## 5.4 Evaluation of RND-GF and the Combination of RND with AT

In this section, we evaluate the performance of RND combined with Gaussian augmentation Fine-tuning, dubbed RNG-GF. In fine-tuning phase, we adopt the cyclic learning rate [44] to achieve superconvergence in 50 epochs. The training detail is shown in Section B.3 of supplementary materials. We adopt the attack failure rate and the average number of queries under the stronger adaptive attack EOT for evaluation. As that in the Section 5.2, we also tune the parameter $\mu$ for attackers and tune $M \in \{1, 5, 10\}$ for EOT to achieve the best attack performance for all attack methods.

The comparison with other defense methods is shown in Table 2, where GF refers to standard fine-tuning with Gaussian augmentation. RND denotes the clean model coupled with random noise defense, and RND-GF indicates the GF model coupled with RND. As shown in the Table 2, RND can significantly boost the defense performance of the Clean Model on both datasets. For example, RNG achieves $20\% \sim 40\%$ improvement against Bandit and SignHunter on both datasets. The proposed

GF can protect the clean accuracy from the larger noise induced by RND. Therefore, we adopt a relatively larger $\nu = 0.05$ for RND-GF. According to our theoretical analysis, larger $\nu$ will lead to better defense performance. Experimental results also show that RNG-GF significantly improves the defense performance under all attack methods while maintaining good clean accuracy compared with RND. For example, RNG-GF achieves $40\% \sim 50\%$ and $20\% \sim 30\%$ improvements against ZO and search-based attacks on CIFAR-10. Compared with RSE, PNI, and FD, RND-GF achieves a higher failure rate against Bandit, SimBA, and ECO and leads to much more queries on other attacks. Besides, it also maintains a much better clean accuracy and lower training cost.

Without any extra modification, RND can be easily combined with many existing defense methods. We adopt the pre-trained AT WideNet-28 ($\ell_\infty$) model [22] and pre-trained AT ResNet-50 ($\ell_\infty$) model in Robustness Library [17] on CIFAR-10 and ImageNet, respectively. As shown in Figure 1 (d), AT models are less affected by adding noise like GF models. So we can adopt the larger noise ($\nu = 0.05$) in inference time. Combining AT with RND, RND-AT significantly improves the robustness against all attacks and achieves the best performance among all methods. Based on AT models, RND-AT achieves $23.1\%$ and $22.7\%$ improvements against Square attack on CIFAR-10 and ImageNet.

### 5.5 Discussion and Limitations

As shown in Section 3.1, we focus on the score-based attacks. **The analysis and evaluation of RND against decision-based attacks are shown in Section C of supplementary materials**. Besides, as we mainly focus on practical query-based black-box attacks where the models and the training datasets are unknown to the attackers, we don't cover the attacks utilizing the transferability from surrogate to target models. These methods usually assume that the surrogate models are trained on the same training set with the target model. It is difficult to obtain the training set behind the target model in real scenarios. Meanwhile, there have been some defense strategies developed for transfer-based attacks [38, 50, 53]. It is interesting to explore the combination of RND and these works towards better defense performance against transfer-based attacks. We leave it to future works.

## 6 Conclusion

In this work, we study a lightweight black-box defense, dubbed Random Noise Defense (RND) for query-based black-box attacks, which is realized by adding proper Gaussian noise to each query. We give the first theoretical analysis of the effectiveness of RND against both standard and adaptive query-based black-box attacks. Based on our analysis, we further propose RND-GF ,which combines RND with Gaussian augmentation Fine-tuning to obtain a better trade-off between clean accuracy and robustness. Extensive experiments verify our theoretical analysis and demonstrate the effectiveness of the RND-GF against several state-of-the-art query-based attacks. Without any extra modification, RND can easily combine with the existing defense methods, such as AT. We further demonstrate that when combined with RND, RND-AT can significantly boost the adversarial robustness. Given that RND is very simple and effective, we recommend it should become a baseline to evaluate new query-based black-box attack methods.

## Acknowledgment

We want to thank Jiancong Xiao, Ziniu Li, Congliang Chen, Jiawei Zhang, and Yingru Li for helpful discussions. We want to thank the anonymous reviewers for their valuable suggestions and comments. Baoyuan Wu and Zeyu Qin are partially supported by the Natural Science Foundation of China under grant No.62076213, the university development fund of the Chinese University of Hong Kong, Shenzhen under grant No.01001810, the special project fund of Shenzhen Research Institute of Big Data under grant No.T00120210003, and Tencent AI Lab Rhino-Bird Focused Research Program under grant No. JR202123. Hongyuan Zha is partially supported by a grant from Shenzhen Research Institute of Big Data.

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
