# Supplementary Materials of Random Noise Defense against Query-Based Black-Box Attacks

**Zeyu Qin[1]**   **Yanbo Fan[2]**   **Hongyuan Zha[1]**   **Baoyuan Wu[1][†]**
[1]School of Data Science, Shenzhen Research Institute of Big Data,
The Chinese University of Hong Kong, Shenzhen
[2]Tencent AI Lab
zeyuqin@link.cuhk.edu.cn, fanyanbo0124@gmail.com,
zhahy@cuhk.edu.cn, wubaoyuan@cuhk.edu.cn

In this supplementary document, we provide additional materials to supplement our main submission. In Section A, we talk about the societal impacts of our work In Section B, we provide detailed experimental settings as well as further evaluation results on CIFAR-10 and ImageNet. We also provide the comparison with input transformation-based defense methods. In Section D, we give the proofs *w.r.t.* Theorem 1 of the main submission. In Section E, we give the proofs *w.r.t.* Theorem 2 of the main submission. The proofs of Theorem 3 are given in Section F.

In Section C, we provide the analysis and evaluation of decision-based attacks.

## A    Societal Impacts

Deep neural networks (DNNs) have been successfully applied in many safety-critical tasks, such as autonomous driving, face recognition and verification, *etc*. And adversarial samples have posed a serious threat to machine learning systems. For real-world applications, the DNN model as well as the training dataset, are often hidden from users. Instead, only the model feedback for each query (*e.g.*, labels or confidence scores) are accessible. In this case, the product providers mainly face the severe threats from query-based black-box attacks, which don't require any knowledge about the attacked models. In this work, we study a lightweight defense method RND against query-based black-box attacks. We conduct the detailed theoretical and empirical analysis of performance of RND against query-based black-box attacks. Extensive experiments verify our theoretical analysis and show the effectiveness of our defense methods against several state-of-the-art query-based attacks. Besides, RND can be directly combined with any off-the-shelf models and other defense strategies.

Therefore, RND is efficient and effective to defend query-based black-box attacks in real scenarios. Without any extra modification, RND can be add to deployed machine learning systems to boost the adversarial robustness.

---

[†]Corresponding author.

35th Conference on Neural Information Processing Systems (NeurIPS 2021).

# B  Experiments

We conduct all experiments on 2 Nvidia-V100 GPU. And we run all experiments 3 times and average all results over 3 random seeds.

RND is very easy to implemented. The defenders add one line code in the Pytorch framework [22], x = x + noise_size * torch.randn_like(x). noise_size is the $\nu$ used in main submission. In inference time, the defenders adopt this method after image transformations and before normalization.

## B.1  Implementation Details of Black-Box Methods

The CIFAR-10 dataset contains 60,000 32×32 color images in 10 different classes, which can be seperated into 50,000 training samples and 10,000 testing samples. These two datasets are licensed under MIT. ImageNet contains 1,000 classes with 1.28 million images for training and 50k images for validation. Imagenet is licensed under Custom (non-commercial).

We evaluate seven mainstreamed query-based attack methods: NES [16], ZOsignSGD [18], Bandit Prior [17], SimBA [13], SignHunter [1], ECO [20], and Square attack [2]. For NES, ZOsignSGD, Bandit Prior, SimBA, and SignHunter, we adopt the source code provided by the authors of SignHunter [1]. For Square attack, we use the source code provided by the authers [2]. For ECO, we adopt the source code provided by the authors [3]. For Signhunter, we only evaluate it under the $\ell_\infty$ attack, since its performance under the $\ell_2$ attack is worse than NES [1]. SimBA is only designed for $\ell_2$ attack. For all attacks, we adopt the hyperparameters recommended by the corresponding papers, which are also shown in Table 1-7. For the evaluation on ImageNet, we use the randomly sampled 1000 images provided by [15][4].

## B.2  Implementation Details of Compared Methods

We compare our methods with adversarial training (AT) [11, 12], Feature Denoise [26], RSE [19] and PNI [14]. For AT model, we adopt the pre-trained WideResNet-28-10 AT model [5] in CIFAR-10. It utilizes the extra unlabeled data. For ImageNet, we adopt the pre-trained ResNet-50 AT model in Robustness Library [6]. It's trained with $4/255\ \ell_\infty$ attack. For Feature Denoise, we adopt the pre-trained ResNet-152 model [7]. For RSE and PNI, we use source code [8] [9] provided by the authors to train the corresponding WideResNet-28-10 models. For standard training VGG model and WideResNet models in CIFAR-10, we use the codes [10] [11] to train them. For used models on ImageNet, we adopt the pre-trained checkpoints provided by *torchvision*.

## B.3  Implementation Details of Gaussian Augmentation Fine-tuning

For GF model, on CIFAR-10, we fine-tune WideResNet-28-10 model with random Gaussian noise sampled from $\mathcal{N}(\mathbf{0}, 0.1\mathbf{I})$. We use the SGD optimizer with momentum 0.9 and weight decay $5*10^{-4}$ We used a variation of the learning rate schedule from [24] to achieve superconvergence in 50 epochs, which is piecewise linear from 0 to 0.001 over the first 20 epochs, down to 0.0005 over the next 20 epochs, and finally back down to 0 in the last 10 epochs.

On ImageNet, [23] released the ResNet-50 model fine-tuned with Gaussian noise sampled from $\mathcal{N}(\mathbf{0}, 0.5\mathbf{I})$ and we directly adopt it.

---

[1] https://github.com/ash-aldujaili/blackbox-adv-examples-signhunter
[2] https://github.com/max-andr/square-attack
[3] https://github.com/snu-mllab/parsimonious-blackbox-attack
[4] https://github.com/TransEmbedBA/TREMBA
[5] https://github.com/deepmind/deepmind-research/tree/master/adversarial_robustness
[6] https://github.com/MadryLab/robustness
[7] https://github.com/facebookresearch/ImageNet-Adversarial-Training
[8] https://github.com/xuanqing94/BayesianDefense
[9] https://github.com/elliothe/CVPR_2019_PNI
[10] https://github.com/kuangliu/pytorch-cifar
[11] https://github.com/DengpanFu/RobustAdversarialNetwork

Table 1: Hyperparameters setup for NES

| Hyperparameter | CIFAR-10 | | ImageNet | |
| --- | --- | --- | --- | --- |
| | $\ell_\infty$ | $\ell_2$ | $\ell_\infty$ | $\ell_2$ |
| $\eta$ (learning rate) | 0.01 | 0.25 | 0.005 | 1 |
| $q$ (number of finite difference estimations per step) | 30 | | 60 | |

Table 2: Hyperparameters setup for ZOsignSGD (ZS)

| Hyperparameter | CIFAR-10 | | ImageNet | |
| --- | --- | --- | --- | --- |
| | $\ell_\infty$ | $\ell_2$ | $\ell_\infty$ | $\ell_2$ |
| $\eta$ (learning rate) | 0.01 | 0.20 | 0.005 | 0.1 |
| $q$ (number of finite difference estimations per step) | 30 | | 60 | |

Table 3: Hyperparameters setup for Bandit Prior

| Hyperparameter | CIFAR-10 | | ImageNet | |
| --- | --- | --- | --- | --- |
| | $\ell_\infty$ | $\ell_2$ | $\ell_\infty$ | $\ell_2$ |
| $\eta$ (learning rate) | 0.01 | 0.25 | 0.01 | 0.5 |
| $h$ (OCO learning rate) | 0.1 | | 0.0001 | |
| $\delta$ (Bandit exploration) | 0.1 | | 0.1 | |
| Tile Size (Data-dependent prior) | 20 | | 50 | |

Table 4: Hyperparameters setup for SimBA

| Hyperparameter | CIFAR-10 | ImageNet |
| --- | --- | --- |
| | $\ell_2$ | $\ell_2$ |
| $\eta$ (step size) | 0.2 | 0.2 |

Table 5: Hyperparameters setup for SignHunter

| Hyperparameter | CIFAR-10 | ImageNet |
| --- | --- | --- |
| | $\ell_\infty$ | $\ell_\infty$ |
| $\eta$ (step size) | 0.05 | 0.05 |

Table 6: Hyperparameters setup for Square Attack (Square)

| Hyperparameter | CIFAR-10 | | ImageNet | |
| --- | --- | --- | --- | --- |
| | $\ell_\infty$ | $\ell_2$ | $\ell_\infty$ | $\ell_2$ |
| $\mu$ (Fraction of Pixel Changed) | $0.05 \sim 0.5$ | | $0.05 \sim 0.5$ | |

Table 7: Hyperparameters setup for ECO

| Hyperparameter | CIFAR-10 | ImageNet |
| --- | --- | --- |
| | $\ell_\infty$ | $\ell_\infty$ |
| block size | 4 | 16 |
| block batch size | 64 | 64 |

## B.4 Additional experimental results of Section 5.2

**Experimental results of RND against Query-based $\ell_2$ Attacks.** Here we provide the defense performance of RND with various settings of $\mu$ and $\nu$ against query-based black-box $\ell_2$ attack. We adopt the same parameter setting as $\ell_\infty$ attack. Figure 1 (a-d) and Figure 2 (a-d) present the defense performance of RND with VGG-16 and WideResNet-16 against NES, ZS, Bandit and Square attack on CIFAR-10, respectively. The experimental results on ImageNet are shown in Figure 3 (a-d). From the results, **we have similar observations with that against $\ell_\infty$ attack**: 1) When $\nu = 0.0$ (i.e., without RND), the attack failure rate of all $\ell_2$ attack methods is very low at all values of $\mu$ and $\nu$, which verifies the poor robustness of the standard model against the query-based attacks. 2) For RND with $\nu > 0$, the attack failure rate of all attack methods generally increases as the value of $\frac{\nu}{\mu}$ increases. While for a certain value of $\mu$ (0.0001 or 0.001 for ZO attacks and 0.1 or 0.3 for Square attack), the attack failure rate increases as the value of $\nu$ increases. These verify our theoretical analysis that the ratio of $\frac{\nu}{\mu}$ determines the upper bound of the convergence rate of query-based ZO attacks and the probability of changing the sign. The larger the value of $\frac{\nu}{\mu}$ results in the poorer attack performance under the query-limited settings. The evaluations in terms of average and median number of queries of successful attacks are shown in Table 8-16, respectively. For brevity, we only report the numerical results with $\mu \in \{0.0001, 0.001, 0.01\}$ for ZO attacks and $\mu \in \{0.1, 0.3, 0.5\}$ for Square attack. And, we set $\nu \in \{0.0, 0.01, 0.02\}$. From the table, we can see that the average query number and the median query number of successful attacks increases as the ratio of $\frac{\nu}{\mu}$ increases. Thus RND can improve the defense performance by increasing the attack failure rate and reducing the query efficiency of black-box attacks. The results of RND with $\mu = 0.0001, 0.0005, 0.001, 0.005, 0.01$ under $\ell_\infty$-attack on ImageNet is also provided in Figure 4.

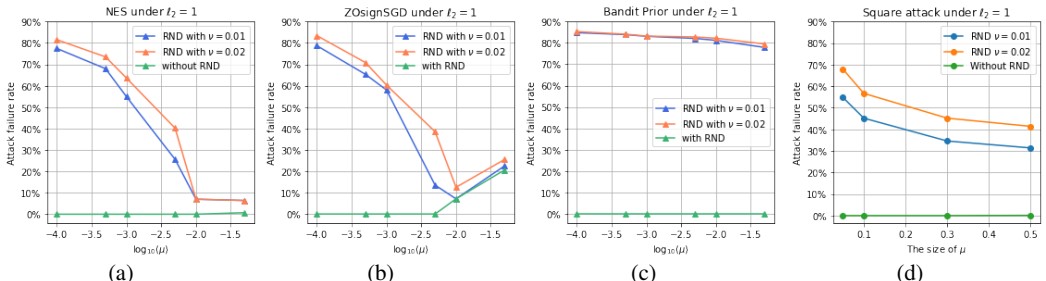

Figure 1: Attack failure rate (%) of query-based attacks on WideResNet-16 and CIFAR-10 under different values of $\mu$ and $\nu$. We adopt logarithm scale in subplot (a-c) for better illustration.

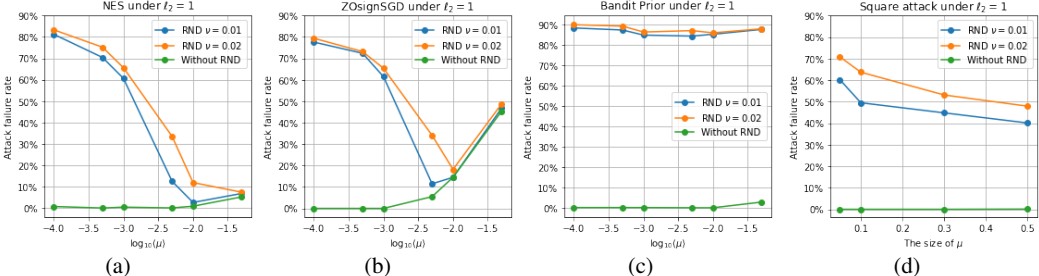

Figure 2: Attack failure rate (%) of query-based attacks on VGG-16 and CIFAR-10 under different values of $\mu$ and $\nu$. We adopt logarithm scale in subplot (a-c) for better illustration.

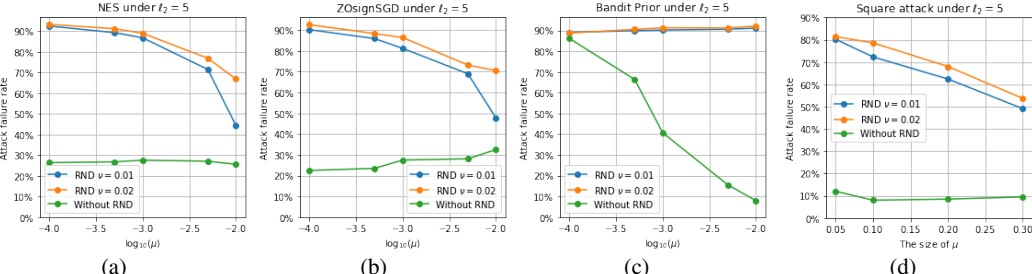

Figure 3: Attack failure rate (%) of query-based attacks on Inception v3 and ImageNet under different values of $\mu$ and $\nu$. We adopt logarithm scale in subplot (a-c) for better illustration.

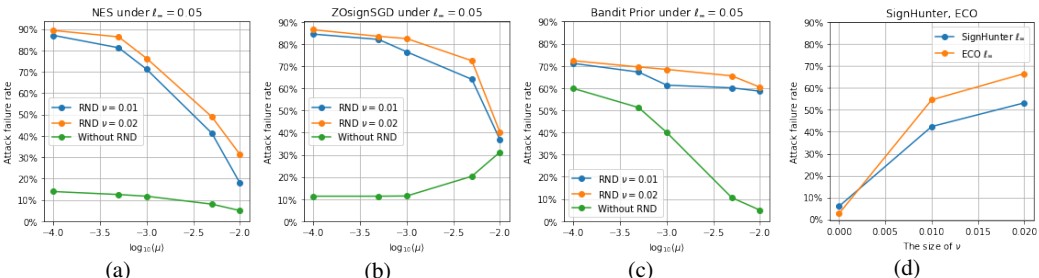

Figure 4: Attack failure rate (%) of query-based attacks on Inception v3 and ImageNet under different values of $\mu$ and $\nu$. We adopt logarithm scale in subplot (a-c) for better illustration.

Table 8: The experimental results of NES, ZS, and Bandit attack on CIFAR-10, VGG-16 model. Each value in this table means the average number of query of successful attack, the median of query, and failure rate $\in [0, 1]$.

| Methods $\mu/\nu$ | $10^{-4}/0.0$ | $10^{-4}/0.01$ | $10^{-4}/0.02$ | $10^{-3}/0.0$ | $10^{-3}/0.01$ | $10^{-3}/0.02$ | $10^{-2}/0.0$ | $10^{-2}/0.01$ | $10^{-2}/0.02$ |
|---|---|---|---|---|---|---|---|---|---|
| NES ($\ell_\infty$) | 281.0(180.0)/0.0 | 1602.2(570.0)/0.621 | 1660.3(580.0)/0.688 | 280.3(180.0)/0.0 | 1506.0(600.0)/0.437 | 1448.4(510.0)/0.484 | 267.3(180.0)/0.0 | 398.8(220.0)/0.0 | 693.4(330.0)/0.06 |
| NES ($\ell_2$) | 380.2(30.0)/0.0 | 1981.4(810.0)/0.812 | 1974.2(760.0)/0.833 | 378.5(300.0)/0.0 | 1987.7(900.0)/0.606 | 1792.2(690.0)/0.656 | 374.8(270.0)/0.0 | 626.3(360.0)/0.03 | 1009.1(480.0)/0.12 |
| ZOsignSGD ($\ell_\infty$) | 246.8(155.0)/0.0 | 1707.4(620.0)/0.00 | 1729.1(627.0)/0.702 | 257.4(155.0)/0.00 | 1499.6(558.0)/0.485 | 1489.7(496.0)/0.493 | 342.4(155.0)/0.0 | 493.9(240.0)/0.03 | 828.3(372.0)/0.1 |
| ZOsignSGD ($\ell_2$) | 337.0(275.0)/0.00 | 2249.0(868.0)/0.774 | 1785.6(744.0)/0.794 | 340.1(275.0)/0.00 | 1939.0(1147.0)/0.615 | 1851.8(899.0)/0.654 | 452.8(310.0)/0.145 | 706.1(465.0)/0.158 | 1254.7(682.0)/0.182 |
| Bandit ($\ell_\infty$) | 117.7(56.0)/0.00 | 402.5(20.0)/0.744 | 410.8(20.0)/0.768 | 115.8(52.0)/0.0 | 382.5(18.0)/0.677 | 436.4(21.0)/0.696 | 111.3(54.0)/0.0 | 168.4(36.0)/0.58 | 183.6(32.0)/0.622 |
| Bandit ($\ell_2$) | 394.1(186.0)/0.00 | 987.2(820.0)/0.885 | 1000.4(80.0)/0.901 | 389.1(190.0)/0.0 | 981.9(82.0)/0.850 | 967.2(83.0)/0.865 | 393.0(182.0)/0.0 | 919.6(102.0)/0.854 | 912.4(100.0)/0.861 |

Table 9: The experimental results of Square attack on CIFAR-10, VGG-16 model. Each value in this table means the average number of query of successful attack, the median of query, and failure rate $\in [0, 1]$.

| Methods $\mu/\nu$ | 0.1/0.0 | 0.1/0.01 | 0.1/0.02 | 0.3/0.0 | 0.3/0.01 | 0.3/0.02 | 0.5/0.0 | 0.5/0.01 | 0.5/0.02 |
|---|---|---|---|---|---|---|---|---|---|
| Square ($\ell_\infty$) | 64.7(26.0)/0.0 | 196.3(25.0)/0.152 | 194.9(12.0)/0.215 | 101.6(37.0)/0.0 | 203.9(26.0)/0.08 | 235.1(19.0)/0.134 | 150.9(60.0)/0.0 | 194.2(20.0)/0.04 | 261.4(33.0)/0.105 |
| Square ($\ell_2$) | 407.1(160.0)/0.0 | 381.4(40.0)/0.495 | 695.1(46.0)/0.637 | 470.4(199.0)/0.0 | 687.7(48.0)/0.448 | 552.9(54.0)/0.531 | 567.8(239.0)/0.0 | 626.3(60.0)/0.40 | 544.1(67.0)/0.479 |

Table 10: The experimental results of SignHunter, SimBA, and ECO on CIFAR-10, VGG-16 model. Each value in this table means the average number of query of successful attack, the median of query, and failure rate $\in [0, 1]$.

| Methods $\nu$ | 0.0 | 0.01 | 0.02 |
|---|---|---|---|
| SignHunter ($\ell_\infty$) | 106.8(570.0)/0.0 | 372.3(64.0)/0.208 | 459.2(55.0)/0.367 |
| SimBA ($\ell_2$) | 308.0(136.0)/0.0 | 1707.4(623.0)/0.473 | 1352.8(172.0)/0.650 |
| ECO ($\ell_\infty$) | 207.5(146.0)/0.0 | 964.6(256.0)/0.609 | 904.3(232.0)/0.720 |

Table 11: The experimental results of NES, ZS, and Bandit attack on CIFAR-10, WideNet-16 model. Each value in this table means the average number of query of successful attack, the median of query, and failure rate $\in [0, 1]$.

| Methods $\mu/\nu$ | $10^{-4}/0.0$ | $10^{-4}/0.01$ | $10^{-4}/0.02$ | $10^{-3}/0.0$ | $10^{-3}/0.01$ | $10^{-3}/0.02$ | $10^{-2}/0.0$ | $10^{-2}/0.01$ | $10^{-2}/0.02$ |
|---|---|---|---|---|---|---|---|---|---|
| NES ($\ell_\infty$) | 265.5(180.0)/0.00 | 1692.5(480.0)/0.606 | 1702.0(570.0)/0.645 | 250.4(180.0)/0.00 | 1626.0(630.0)/0.389 | 1572.8(570.0)/0.434 | 264.8(180.0)/0.0 | 352.6(210.0)/0.0 | 731.9(330.0)/0.03 |
| NES ($\ell_2$) | 368.4(270.0)/0.0 | 1942.9(720.0)/0.774 | 1735.8(800.0)/0.815 | 374.9(270.0)/0.0 | 2046.5(990.0)/0.549 | 1715.1(750.0)/0.636 | 304.4(240.0)/0.0 | 550.0(330.0)/0.0 | 1115.7(510.0)/0.07 |
| ZOsignSGD ($\ell_\infty$) | 188.4(124.0)/0.0 | 1657.7(496.0)/0.591 | 1638.0(523.0)/0.642 | 191.0(124.0)/0.0 | 1477.0(620.0)/0.424 | 1447.3(496.0)/0.453 | 190.1(124.0)/0.0 | 436.7(217.0)/0.012 | 747.5(341.0)/0.065 |
| ZOsignSGD ($\ell_2$) | 295.3(248.0)/0.0 | 2181.4(868.0)/0.787 | 2254.5(850.0)/0.834 | 296.1(248.0)/0.0 | 2156.6(1209.0)/0.580 | 2136.2(1312.0)/0.603 | 580(310.0)/0.008 | 772.8(434.0)/0.071 | 1237.4(744.0)/0.125 |
| Bandit ($\ell_\infty$) | 92.51(44.0)/0.0 | 108.1(30.0)/0.647 | 124.7(31.0)/0.672 | 90.76(42.0)/0.0 | 382.5(40.0)/0.602 | 401.6(39.0)/0.625 | 91.98(42.0)/0.0 | 214.8(43.0)/0.461 | 345.7(43.0)/0.486 |
| Bandit ($\ell_2$) | 281.6(158.0)/0.0 | 1592.(160.0)/0.848 | 1611.3(152.0)/0.853 | 282.3(158.0)/0.0 | 1595.8(160.0)/0.831 | 1621.3(160.0)/0.832 | 275.12(158.0)/0.0 | 973.0(84.0)/0.812 | 1125.0(78.0)/0.822 |

Table 12: The experimental results of Square attack on CIFAR-10, WideNet-16 model. Each value in this table means the average number of query of successful attack, the median of query, and failure rate $\in [0, 1]$.

| Methods $\mu/\nu$ | 0.1/0.0 | 0.1/0.01 | 0.1/0.02 | 0.3/0.0 | 0.3/0.01 | 0.3/0.02 | 0.5/0.0 | 0.5/0.01 | 0.5/0.02 |
|---|---|---|---|---|---|---|---|---|---|
| Square ($\ell_\infty$) | 42.8(19.0)/0.0 | 185.1(27.0)/0.112 | 294.9(22.0)/0.190 | 121.6(27.0)/0.0 | 231.2(26.0)/0.08 | 245.2(19.0)/0.144 | 250.1(60.0)/0.0 | 294.2(23.0)/0.04 | 264.1(35.0)/0.105 |
| Square ($\ell_2$) | 303.1(131.0)/0.0 | 964.3(103.0)/0.452 | 597.8(39.0)//0.567 | 293.4(131.0)/0.0 | 587.2(48.0)/0.346 | 589.4(61.0)/0.456 | 440.1(210.0)/0.0 | 626.3(60.0)/0.314 | 516.4(72.0)/0.414 |

Table 13: The experimental results of SignHunter, SimBA, and ECO on CIFAR-10, WideNet-16 model. Each value in this table means the average number of query of successful attack, the median of query, and failure rate $\in [0, 1]$.

| Methods $\nu$ | 0.0 | 0.01 | 0.02 |
|---|---|---|---|
| SignHunter ($\ell_\infty$) | 95.0(55.0)/0.0 | 415.7(56.0)/0.271 | 651.8(59.0)/0.396 |
| SimBA ($\ell_2$) | 241.2(138.0)/0.0 | 1813.0(702.0)/0.476 | 1359.1(187.0)/0.648 |
| ECO ($\ell_\infty$) | 157.6(76.0)/0.0 | 787.4(236.0)/0.533 | 788.8(218.0)/0.637 |

Table 14: The experimental results of NES, ZS, and Bandit on ImageNet, Inception v3 model. Each value in this table means the average number of query of successful attack, the median of query, and failure rate $\in [0, 1]$.

| Methods $\mu/\nu$ | $10^{-4}$/0.0 | $10^{-4}$/0.01 | $10^{-4}$/0.02 | $10^{-3}$/0.0 | $10^{-3}$/0.01 | $10^{-3}$/0.02 | $10^{-2}$/0.0 | $10^{-2}$/0.01 | $10^{-2}$/0.02 |
|---|---|---|---|---|---|---|---|---|---|
| NES ($\ell_\infty$) | 1678.4(960.0)/0.139 | 3037.5(2040.0)/0.870 | 3010.6(2025.0)/0.893 | 1435.4(900.0)/0.117 | 3402.3(2280.0)/0.711 | 2532.8(1560.0)/0.762 | 1571.2(840.0)/0.05 | 1721.8(1080.0)/0.179 | 2143.4(1380.0)/0.313 |
| NES ($\ell_2$) | 2016.5(1215.0)/0.264 | 1888.9(360.0)/0.925 | 1954.1(345.0)/0.933 | 2021.1(1200.0)/0.276 | 2245.7(1200.0)/0.868 | 1814.2(720.0)/0.890 | 2048.6(1200.0)/0.256 | 1836.0(900.0)/0.444 | 2378.2(1440.0)/0.669 |
| ZOsignSGD ($\ell_\infty$) | 1312.9(671.0)/0.113 | 2232.6(1159.0)/0.845 | 1870.5(1132.0)/0.865 | 1316.7(671.0)/0.114 | 2244.3(1464.0)/0.771 | 2824.1(1586.0)/0.825 | 1544.2(793.0)/0.312 | 1678.5(1203.5)/0.334 | 2355.3(1708.0)/0.373 |
| ZOsignSGD ($\ell_2$) | 1642.6(1350.0)/0.225 | 1389.9(445.0)/0.902 | 1389.9(427.0)/0.926 | 1442.6(1240.0)/0.275 | 1389.9(545.0)/0.812 | 1375.3(490.0)/0.864 | 1945.6(1100.0)/0.325 | 1745.0(985.0)/0.478 | 1245.4(785.0)/0.705 |
| Bandit ($\ell_\infty$) | 696.7(16.0)/0.598 | 114.2(12.0)/0.711 | 122.5(12.0)/0.723 | 1422.6(104.0)/0.40 | 202.1(16.0)/0.613 | 195.4(16.0)/0.684 | 903.1(142.0)/0.050 | 288.4(18.0)/0.586 | 305.6(20.0)/0.604 |
| Bandit ($\ell_2$) | 987.9(118.0)/0.861 | 583.6(60.0)/0.891 | 656.4(42.0)/0.848 | 1483.5(990.0)/0.407 | 615.3(60.0)/0.902 | 578.6(60.0)/0.913 | 1327.2(636.0)/0.088 | 246.5(92.0)/0.911 | 235.2(91.0)/0.921 |

Table 15: The experimental results of Square attack on ImageNet, Inception v3 model. Each value in this table means the average number of query of successful attack, the median of query, and failure rate $\in [0, 1]$.

| Methods $\mu/\nu$ | 0.1/0.0 | 0.1/0.01 | 0.1/0.02 | 0.3/0.0 | 0.3/0.01 | 0.3/0.02 | 0.5/0.0 | 0.5/0.01 | 0.5/0.02 |
|---|---|---|---|---|---|---|---|---|---|
| Square ($\ell_\infty$) | 247.1(23.0)/0.003 | 576.8(19.0)/0.149 | 207.4(18.0)/0.264 | 359.9(40.0)/0.05 | 339.6(25.0)/0.122 | 251.7(17.0)/0.207 | 457.0(66.0)/0.08 | 315.2(28.0)/0.106 | 296.3(16.0)/0.154 |
| Square ($\ell_2$) | 1107.1(310.0)/0.08 | 1030.2(316.0)/0.723 | 997.8(390.0)//0.786 | 1247.5(409.0)/0.09 | 1187.2(480.0)/0.491 | 1089.4(461.0)/0.537 | 1340.1(510.0)/0.11 | 1226.3(460.0)/0.426 | 1236.4(480.0)/0.493 |

Table 16: The Experiment of SignHunter, SimBA, and ECO on ImageNet, Inception v3 model. Each value in this table means the average number of query of successful attack, the median of query, and failure rate $\in [0, 1]$.

| Methods $\nu$ | 0.0 | 0.01 | 0.02 |
|---|---|---|---|
| SignHunter ($\ell_\infty$) | 557.2(108.0)/0.056 | 163.4(50.0)/0.424 | 173.4(49.0)/0.532 |
| SimBA ($\ell_2$) | 2077.4(1398.0)/0.195 | 223.6(21.0)/0.885 | 274.6(19.0)0.891 |
| ECO ($\ell_\infty$) | 853.2(258.0)/0.027 | 1043.2(485.0)/0.432 | 1236.9(691.0)/0.666 |

## B.5 Additional experimental results of Section 5.3

**Experimental results of RND against The Adaptive EOT Attack.** We also evaluate the defense performance of RND against EOT with $\ell_2$ attack. The evaluations with adaptive query budget on

NES and ZS of CIFAR-10 are shown in Table 19 and 20. From the Table, we observe that the attack failure rate decreases as $M$ increases on both datasets. However, the average number of queries of successful attack also greatly increases as M increases, which demonstrates that the adaptive EOT attack increases the attacking success rate with a sacrifice of query efficiency. We also observe that the relative performance improvements induced by EOT under both $\ell_\infty$ and $\ell_2$ attack generally decrease as M increases, especially when $M$ is rather large. For example, for NES with $\ell_\infty$ attack, the relative improvement of $M = 20$ over that of $M = 15$ is only 0.004 in terms of attack failure rate. Yet the average query number of $M = 20$ is 306 higher than that of $M = 15$. These validate our theoretical analysis in Section 4.3 of the main submission that the attack improvement from EOT is limited as M increases.

The experimental results under the fixed query budget on CIFAR-10 and ImageNet are reported in Table 17 and 18, respectively. On these two datasets, the attack failure rate of all attacks generally decreases as M increases. Yet we also observe the similar phenomenon that the relative performance improvements induced by EOT decreases as M increases.

Table 17: The evaluation of EOT with $\ell_2$ attack on CIFAR-10 under the fixed query budget setting. The average number of query of successful attack as well as the attack failure rate are reported. For all attacks, we set $\nu = 0.02$ and set $\mu = 0.001$ for NES, ZS, and Bandit. Each value in this table means the average number of query of successful attack, the median of query, and failure rate $\in [0, 1]$. The higher failure rate, the better defense performance.

| Methods | M=1 | M=5 | M= 10 |
|---------|-----|-----|-------|
| NES | 1792/690/0.656 | 4736/2850/0.598 | 5167/3220/0.523 |
| ZS | 1939/1147/0.615 | 3921/3410/0.578 | 4135/4215/0.541 |
| Bandit | 912/100/0.861 | 662/160/0.782 | 698/193/0.745 |
| Square | 413/42/0.708 | 284/70/0.777 | 263/69/0.815 |
| SimBA | 1353/172/0.650 | 3852/1585/0.467 | 4103/2836/0.396 |

Table 18: The evaluation of EOT with $\ell_2$ attack on ImageNet under the fixed query budget setting. The average number of queries of successful attack as well as the attack failure rate are reported.

| Methods | M=0 | M=5 | M=10 |
|---------|-----|-----|------|
| NES | 1814/0.890 | 4825.6/0.912 | 5801.3/0.925 |
| ZS | 1375/0.864 | 3055/0.887 | 4652/0.861 |
| Bandit | 195/0.684 | 698/0.579 | 873/0.0.553 |
| Square | 160.9/0.822 | 178.0/0.831 | 179.5/0.845 |
| SimBA | 274/0.891 | 468/0.878 | 517/0.869 |

Table 19: The evaluation of EOT with $\ell_\infty$ and $\ell_2$ attack on CIFAR-10 under the adaptive query budget setting. The average number of queries of successful attack as well as the attack failure rate are reported. For all attacks, we choose the same parameter as the Table 17.

| Datasets | Methods | M=1 | M=5 | M=10 | M=15 | M=20 |
|----------|---------|-----|-----|------|------|------|
| CIFAR-10 | NES($\ell_\infty$) | 1448/0.484 | 4078/0.361 | 5763/0.342 | 6126/0.331 | 6342/0.327 |
| | NES($\ell_2$) | 1792/0.656 | 3074/0.513 | 3642/0.456 | 4023/0.432 | 4125/0.429 |
| | ZS($\ell_\infty$) | 1489/0.493 | 3189/0.354 | 5912/0.319 | 6159/0.293 | 7013/0.287 |
| | ZS($\ell_2$) | 1852/0.654 | 4052/0.541 | 4619/0.498 | 4923/0.475 | 4867/0.472 |

Table 20: The evaluation of EOT with $\ell_\infty$ attack on ImageNet under the adaptive query budget setting. The average number of queries of successful attack as well as the attack failure rate are reported. For all attacks, we still set $\nu$ as 0.02 and increase $\mu$ to 0.01.

| Datasets | Methods | M=1 | M=5 | M=10 | M=15 |
|----------|---------|-----|-----|------|------|
| ImageNet | NES($\ell_\infty$) | 0.313 | 0.251 | 0.236 | 0.224 |
| | NES($\ell_2$) | 0.669 | 0.603 | 0.576 | 0.551 |
| | ZS ($\ell_\infty$) | 0.373 | 0.279 | 0.261 | 0.257 |
| | ZS($\ell_2$) | 0.705 | 0.616 | 0.579 | 0.556 |

## B.6 Additional experimental results of Section 5.4

**Experimental results of Compared Methods against $\ell_2$ attack.** We adopt the same experimental setting reported in Section 5.4 of the main submission. The evaluation results on CIFAR-10 and ImageNet are shown in Table 21. As shown in Table 21: 1) the clean model obtains the best clean accuracy while poorest robustness under most attacks; 2) RND can improve the defense performance of clean model on both datasets. Yet the random noise induced by RND will also sacrifice the clean accuracy. 3) GT provides a better protection of clean accuracy under the random noise induced by RND, so we can adopt a relative larger $\nu = 0.05$ for RND-GT towards better defense performance. RNG-GT can significantly improve the defense performance under all attack methods while maintaining a satisfactory clean performance. Similar to the results of $\ell_\infty$ attack in the main submission, compared with RSE, PNI, and Feature Denoise (FD), RND-GF achieves the better defense effect against Bandit, SimBA, and ZS and maintain the much better clean accuracy and low training cost. Combining AT with RND, RND-AT significantly improves the robustness against all attacks and achieves best performance among all methods.

Table 21: The comparison of RND ($\nu = 0.02$), GF, RND-GF ($\nu = 0.05$), AT, RND-AT ($\nu = 0.05$), PNI, RSE, and FD on CIFAR-10 and Imagenet. The average number of queries of successful attack and the attack failure rates are reported. The best and second best attack failure rate under each attack are highlighted in bold and underlined respectively.

| Datasets | Methods | Clean Acc | NES($\ell_2$) | ZS($\ell_2$) | Bandit($\ell_2$) | SimBA($\ell_2$) | Square($\ell_2$) |
|---|---|---|---|---|---|---|---|
| CIFAR-10 (WideNet-28) | Clean Model | **96.60**% | 729.1/0.025 | 967.4/0.224 | 619.0/0.03 | 457.2/0.04 | 631.3/0.03 |
| | RND | 93.60% | 1279.7/0.194 | 1476.1/0.446 | 1624.1/0.762 | 2112.6/0.549 | 1221.5/0.487 |
| | GF | 91.72% | 967.4/0.595 | 826.4/0.645 | 1543.5/0.274 | 1146.8/0.395 | 1626.5/0.328 |
| | RND-GF | 92.40% | 3209.8/0.661 | 2453.2/0.901 | 1362.1/0.838 | 1220.2/0.863 | 1415.3/0.692 |
| | RSE | 84.12% | 1293.6/0.387 | 1367.9/0.391 | 264.5/0.334 | 498.3/0.337 | 599.0/0.231 |
| | PNI | 87.20% | 1457.1/0.812 | 1939.5/0.843 | 897.9/0.861 | 945.0/0.857 | 485.2/0.826 |
| | AT | 89.48% | 1155.4/0.765 | 397.5/0.856 | 2163.2/0.588 | 1523.2/0.635 | 1935.4/0.677 |
| | RND-AT | 87.40% | 3044.4/**0.849** | 2904.0/**0.956** | 1603.5/**0.931** | 1787.4/**0.912** | 1292.8/**0.842** |
| ImageNet (ResNet-50) | Clean Model | **74.90**% | 1335.6/0.03 | 1254.2/0.216 | 856.0/0.0 | 1234.5/0.281 | 621.1/0.01 |
| | RND | 73.00% | 2027.8/0.509 | 2566.1/0.631 | 312.4/0.764 | 825.3/0.612 | 1563.1/0.481 |
| | GF | 74.70% | 1803.7/0.146 | 1902.1/0.194 | 896.3/0.056 | 1417.4/0.112 | 915.7/0.042 |
| | RND-GF | 71.15% | 1542.7/0.760 | 1625.5/0.820 | 511.4/0.875 | 777.2/0.829 | 1130.2/0.625 |
| | FD | 54.20% | 2048.4/0.724 | 709.3/0.812 | 2605.9/0.545 | 2607.9/0.613 | 1539.1/0.482 |
| | AT | 61.60% | 2365.1/0.782 | 639.3/0.912 | 2769.2/0.544 | 2638.2/0.651 | 1404.3/0.528 |
| | RND-AT | 58.15% | 2482.9/**0.926** | 2395.4/**0.937** | 1079.1/**0.935** | 1210.5/**0.953** | 175.0/**0.80** |

## B.7 The Comparison with Input Transformation-based Defense Methods

Apart from the compared randomization-based methods in main submission, we also compare RND with input transformation-based defense methods, R&P [25], JPEG [10], and Bit-Red [27]. We take the comparison on the ImageNet and Inception-v3 model. We adopt the NES, SignHunter, and Square attacks. The maximal query number is 10000. For RND, we set the $\nu$ as 0.02. We report the $\ell_\infty$ attack results. The results *w.r.t.* the standard attack and the adaptive EOT attack with $M = 5$ are given below tables. Compared to these input transformation-based defense methods, RND achieves better results in both clean accuracy and defense performance.

Table 22: The evaluation of RND, R&P, JPEG, and Bit-Red against $\ell_\infty$ attack on ImageNet and Inception-v3 model. We adopt NES, SignHunter, and Square attacks. Each value in this table means the average number of query of successful attack and attack failure rate

| Defense Methods | clean accuracy | NES($\ell$) | SignHunter($\ell$) | Square($\ell$) |
|---|---|---|---|---|
| R&P | 74.6% | 1368.2/0.378 | 345.1/0.223 | 285.8/0.153 |
| Bit-Red | 62.5% | 1548.6/0.092 | 145.4/0.048 | 356.2/0.021 |
| JPEG | 74.2% | 1417.4/0.205 | 156.8/0.146 | 215.3/0.033 |
| RND | 76.6% | 2143.4/0.413 | 173.4/0.532 | 251.7/0.207 |

Table 23: The evaluation of RND, R&P, JPEG, and Bit-Red against $\ell_\infty$ EOT attack ($M = 5$) on ImageNet and Inception-v3 model. We adopt NES, SignHunter, and Square attacks. Each value in this table means the average number of query of successful attack and attack failure rate

| Defense Methods | clean accuracy | NES($\ell$) | SignHunter($\ell$) | Square($\ell$) |
|---|---|---|---|---|
| R&P | 74.6% | 2547.6/0.302 | 545.6/0.148 | 656.2/0.069 |
| Bit-Red | 62.5% | 12682.3/0.031 | 612.3/0.010 | 689.3/0.004 |
| JPEG | 74.2% | 2863.2/0.117 | 485.6/0.094 | 712.5/0.009 |
| RND | 76.6% | 3240.2/0.365 | 336.5/0.456 | 0.9/0.121 |

## C  Analysis and Evaluation of Decision-based Attacks

In this section, we give the analysis of defense effect of RND against decision-based attacks.

For decision-based attacks, the attacker only obtain the classification label by querying the attacked models. As mentioned in main submission, score-based attacks utilize the gradient estimation or random search to find adversarial direction from the benign example to adversarial example with wrong label. However, decision-based attack adopt the different idea to find adversarial examples. Compared with score-based attacks, decision-based attacks first add the large perturbation which make true the found examples in the initial phase can be misclassified. Then, to satisfy the requirement of perturbation size, the attackers need to reduce the distance between found example and the benign example and still have to make sure that the found example can be misclassified. Therefore, the decision-based attacks conduct attacks from adversarial example with large perturbation to benign examples. The found adversarial examples will eventually fall near the boundary [3, 4, 5, 6, 7, 9].

Decision-based attacks can be also separated into gradient estimation and random search-based attacks. Gradient estimation attacks contain Sign-OPT attack [9] and OPT attack [7], and HopSkipJumpAttack (HSJA) [4]. search-based attacks contain boundary attack [3], Sign Flip [6] and RayS [5]. To find the adversarial examples near the decision boundary, binary search is widely used in these two methods. Then, **gradient estimation [4, 7, 9] or random search [5, 6] are conducted to find attack direction near the decision boundary**. Therefore, our analysis about ZO attacks and random search still apply to decision attacks.

We conduct the evaluation of RND against decision-based attacks to verify our analysis. We evaluate Sign-OPT attack, HSJA, Sign Flip, and Rays, because these four attack methods show the better attack performance. We report the performance of RND on WideNet-28-10 and CIFAR-10. We also utilize the Gaussian augmentation fine-tuning to fine the WideNet-28. The work in [4, 5, 6] adopt the fixed size schedule. Therefore, we only tune the noise size $\mu$ of Sign-OPT attack like NES [16], zosignsgd [18], Bandit [17], and Square attack [2]. The clean accuracy under different noise has shown in main submission. The experiments results are shown in next tables.

The results in below tables show that: the attack failure rate of attack methods generally increases as the value of $\frac{\nu}{\mu}$ increases. These collaborate our theoretical analysis that **the ratio of $\frac{\nu}{\mu}$ determines the probability of changing sign and the convergence rate of ZO attacks.** The larger $\frac{\nu}{\mu}$, the higher the probability of changing the sign and the convergence error of ZO attacks, which results in the poor attack performance of decision-based attacks under the query-limited settings.

As shown in the below tables, RND can significant boost the defense performance of the Clean Model against the decision-based attacks. Based on GF, RNG-GF further improves the defense performance under all attack methods while maintaining the good clean accuracy.

Table 24: The experimental results of Sign-OPT $\ell_\infty$ attack on CIFAR-10, WideNet-28 model. Each value in this table means the average number of query of successful attack, and failure rate $\in [0, 1]$.

| Methods $\mu/\nu$ | 0.01/0.0 | 0.01/0.01 | 0.01/0.02 | 0.05/0.0 | 0.05/0.01 | 0.05/0.02 | 0.1/0.0 | 0.1/0.01 | 0.1/0.02 |
|---|---|---|---|---|---|---|---|---|---|
| Sign-OPT (RND) | 1999.5/0.028 | 1205.2/0.929 | 1008.7/0.935 | 2153.5/0.034 | 1235.2/0.890 | 1428.6/0.926 | 2477.6/0.044 | 1820.3/0.870 | 1959.6/0.921 |

Table 25: The experimental results of HSJA, Sing Flip, and RayS attacks under $\ell_\infty$ norm on CIFAR-10, WideNet-28 model. Each value in this table means the average number of query of successful attack, and failure rate $\in [0, 1]$.

| Methods \ $\nu$ | 0.0 | 0.01 | 0.02 |
|---|---|---|---|
| HSJA ($\ell_\infty$) | 977.7/ 0.002 | 2792.6/0.593 | 3370.8/0.623 |
| Sign Flip ($\ell_\infty$) | 222.1/0.0 | 1095.0/0.360 | 564.4/0.540 |
| RayS ($\ell_\infty$) | 685.9/0.0 | 998.7/0.06 | 865.6/0.210 |

Table 26: The experimental results of Sign-OPT $\ell_\infty$ attack on CIFAR-10, WideNet-28 GF model. Each value in this table means the average number of query of successful attack, and failure rate $\in [0, 1]$.

| Methods \ $\mu/\nu$ | 0.01/0.0 | 0.01/0.02 | 0.01/0.05 | 0.05/0.0 | 0.05/0.02 | 0.05/0.05 | 0.1/0.0 | 0.1/0.02 | 0.1/0.05 |
|---|---|---|---|---|---|---|---|---|---|
| Sign-OPT (RND-GF) | 3257.6/0.651 | 2217.3/0.783 | 541.6/0.961 | 2950.4/0.670 | 2017.1/0.735 | 641.5/0.961 | 3969.1/0.682 | 1516.4/0.716 | 465.2/0.938 |

Table 27: The experimental results of HSJA, Sing Flip, and RayS attacks under $\ell_\infty$ norm on CIFAR-10, WideNet-28 GF model. Each value in this table means the average number of query of successful attack, and failure rate $\in [0, 1]$.

| Methods \ $\nu$ | 0.0 | 0.02 | 0.05 |
|---|---|---|---|
| HSJA ($\ell_\infty$) | 4629.8/0.11 | 4878.5/0.795 | 6177.2/0.952 |
| Sign Flip ($\ell_\infty$) | 2337.9/ 0.115 | 3134.9/0.736 | 1565.6/0.961 |
| RayS ($\ell_\infty$) | 1303.5/0.034 | 314.6/0.324 | 572.8/0.564 |

Table 28: The experimental results of Sign-OPT $\ell_\infty$ attack on ImageNet, Inception v3 model. Each value in this table means the average number of query of successful attack, and failure rate $\in [0, 1]$.

| Methods \ $\mu/\nu$ | 0.01/0.0 | 0.01/0.01 | 0.01/0.02 | 0.05/0.0 | 0.05/0.01 | 0.05/0.02 | 0.1/0.0 | 0.1/0.01 | 0.1/0.02 |
|---|---|---|---|---|---|---|---|---|---|
| Sign-OPT (RND) | 6901.5/0.174 | 2512.6/0.912 | 1903.5/0.976 | 6432.6/0.243 | 2465.6/0.884 | 1963.1/0.931 | 7602.5/0.391 | 2278.6/0.842 | 2603.4/0.926 |

Table 29: The experimental results of HSJA, Sing Flip, and RayS attacks under $\ell_\infty$ norm on ImageNet, Inception v3 model. Each value in this table means the average number of query of successful attack, and failure rate $\in [0, 1]$.

| Methods \ $\nu$ | 0.0 | 0.01 | 0.02 |
|---|---|---|---|
| HSJA ($\ell_\infty$) | 2912.9/ 0.150 | 3289.2/0.791 | 3370.8/0.879 |
| Sign Flip ($\ell_\infty$) | 2216.1/0.105 | 1485.2/0.756 | 1400.6/0.864 |
| RayS ($\ell_\infty$) | 1321.5/0.067 | 947.2/0.284 | 787.2/0.326 |

# D Proof of Section 4.2.1

We first give some function properties we will use in next sections:

**Definition 1.** *The Gaussian-Smoothing function corresponding to $f(\boldsymbol{x})$ with $\alpha > 0$ is defined as follows*

$$f_\alpha(\boldsymbol{x}) = \frac{1}{(2\pi)^{d/2}} \int f(\boldsymbol{x} + \alpha \boldsymbol{a}) \cdot e^{-\frac{1}{2}\|\boldsymbol{a}\|_2^2} \, d\boldsymbol{a}. \tag{1}$$

Here, $\alpha \geq 0$ is the smoothing parameter. And if $f$ is convex and the subgradient $\boldsymbol{g} \in \partial f(\boldsymbol{x})$, then

$$f_\mu(\boldsymbol{x}) \geq \frac{1}{(2\pi)^{d/2}[\det\boldsymbol{\Sigma}]^{d/2}} \int [f(\boldsymbol{x}) + \mu\langle\boldsymbol{g},\boldsymbol{u}\rangle]e^{-\frac{1}{2}\|\boldsymbol{u}\|^2} \, d\boldsymbol{u} = f(\boldsymbol{x})$$

If $f \in C^{0,0}$, then $f_\mu \in C^{0,0}$ and $L_0(f_\mu) \leq L_0(f)$. Indeed, for all $x, y \in R^d$ we have

$$|f_\mu(x) - f_\mu(y)| \leq \frac{1}{(2\pi)^{d/2}[\det\boldsymbol{\Sigma}]^{d/2}} \int |f(x + \mu u) - f(y + \mu u)|e^{-\frac{1}{2}\|u\|^2} \, du$$

$$\leq L_0(f)\|x - y\|$$

If $f \in C^{1,1}$, then $f_\mu \in C^{1,1}$ and $L_1(f_\mu) \leq L_1(f)$ :

$$\|\nabla f_\mu(x) - \nabla f_\mu(y)\| \leq \frac{1}{(2\pi)^{d/2}[\det\boldsymbol{\Sigma}]^{d/2}} \int_E \|\nabla f(x + \mu u) - \nabla f(y + \mu u)\|e^{-\frac{1}{2}\|u\|^2} \, du$$

$$\leq L_1(f)\|x - y\|$$

For the gradients of $f_\mu(x)$,

$$f_\mu(\boldsymbol{x}) = \frac{1}{\mu^{d+1}(2\pi)^{d/2}[\det\boldsymbol{\Sigma}]^{d/2}} \int f(\boldsymbol{y})e^{-\frac{1}{2\mu^2}\|\boldsymbol{y}-\boldsymbol{x}\|^2} \, d\boldsymbol{y}$$

$$\nabla f_\mu(\boldsymbol{x}) = \frac{1}{\mu^{d+3}(2\pi)^{d/2}[\det\boldsymbol{\Sigma}]^{d/2}} \int f(\boldsymbol{y})e^{-\frac{1}{2\mu^2}\|\boldsymbol{y}-\boldsymbol{x}\|^2}(\boldsymbol{y}-\boldsymbol{x})d\boldsymbol{y}$$

$$= \frac{1}{\mu(2\pi)^{d/2}[\det\boldsymbol{\Sigma}]^{d/2}} \int f(\boldsymbol{x} + \mu\boldsymbol{u})e^{-\frac{1}{2}\|\boldsymbol{u}\|^2}\boldsymbol{u} \, d\boldsymbol{u}$$

$$= \frac{1}{(2\pi)^{d/2}[\det\boldsymbol{\Sigma}]^{d/2}} \int \frac{f(\boldsymbol{x} + \mu\boldsymbol{u}) - f(\boldsymbol{x})}{\mu}e^{-\frac{1}{2}\|\boldsymbol{u}\|^2}\boldsymbol{u} \, d\boldsymbol{u}$$

We can see that the gradient estimator $g_\mu$ is the unbiased estimator of $\nabla f_\mu(\boldsymbol{x})$. Denote by $f'(\boldsymbol{x}, \boldsymbol{u})$ the directional derivative of $f$ at point $\boldsymbol{x}$ along direction $\boldsymbol{u}$:

$$f'(\boldsymbol{x}, \boldsymbol{u}) = \langle\nabla f(\boldsymbol{x}), \boldsymbol{u}\rangle = \lim_{\mu\downarrow 0}\frac{1}{\mu}[f(\boldsymbol{x} + \mu\boldsymbol{u}) - f(\boldsymbol{x})]$$

$$\nabla f(\boldsymbol{x}) = \frac{1}{(2\pi)^{d/2}[\det\boldsymbol{\Sigma}]^{d/2}} \int f'(\boldsymbol{x}, \boldsymbol{u})e^{-\frac{1}{2}\|\boldsymbol{u}\|^2}\boldsymbol{u} \, d\boldsymbol{u}$$

Next, we give some essential lemmas. The complete proofs are shown in [21].

**Lemma 1.** *Let $f$ be the Lipschitz-continuous function, $|f(\boldsymbol{y}) - f(\boldsymbol{x})| \leq L_0(f)\|\boldsymbol{y} - \boldsymbol{x}\|$. Then*

$$L_1(f_\mu) = \frac{d^{\frac{1}{2}}}{\mu}L_0(f)$$

**Lemma 2.** *For the smoothed version $f_\mu$ of $f$, if both of them has Lipschitz-continuous gradient, then*

$$L_1(f_\mu) \leq L_1(f)$$

And we define the $p$-order moment of normal distribution as $M_p$. We need the upper bound for moment of standard Gaussian distribution.

**Lemma 3.** *For $p \in [0, 2]$, we have*

$$M_p \leq d^{p/2}$$

*If $p \geq 2$, then we have two-side bounds*

$$d^{p/2} \leq M_p \leq (p + d)^{p/2}$$

## D.1 The General Non-Convex Case

Recalling the optimization problem for attacker:

$$\min_{\boldsymbol{x}_{adv}} f(\boldsymbol{x}_{adv})$$
$$St. \|\boldsymbol{x}_{adv} - \boldsymbol{x}\|_p \leq R$$

The gradient estimator $g(\boldsymbol{x})$ in ZO attacks becomes

$$g_{\mu,\nu}(\boldsymbol{x}) = \frac{f(\boldsymbol{x} + \mu\boldsymbol{u} + \nu\boldsymbol{v}_1) - f(\boldsymbol{x} + \nu\boldsymbol{v}_2)}{\mu}\boldsymbol{u} \tag{2}$$

Here, **we use Euclidean norm in our all theoretical analysis**.

We firstly define

$$f_{\mu,\nu}(\boldsymbol{x}) = \frac{1}{(2\pi)^{d/2}} \int f_\nu(\boldsymbol{x} + \mu\boldsymbol{u})\mathrm{e}^{-\frac{1}{2}\|\boldsymbol{u}\|^2} \, \mathrm{d}\boldsymbol{u}$$

which is the smoothing version of $f_\nu(\boldsymbol{x})$.

We denote the sequence of standard Gaussian noises added by the attacker as $\mathcal{U}_t = \{\boldsymbol{u}_0, \boldsymbol{u}_1, \ldots, \boldsymbol{u}_t\}$. The sequence of standard Gaussian noises added by the defender is denoted as $\mathcal{V}_t = \{\boldsymbol{v}_{01}, \boldsymbol{v}_{02}, \ldots, \boldsymbol{v}_{t1}, \boldsymbol{v}_{t2}\}$. The sequential solutions generated are denoted as $\{\boldsymbol{x}_0, \boldsymbol{x}_1, \ldots, \boldsymbol{x}_Q\}$, and the benign example $\boldsymbol{x}$ is used as the initial solution, $\boldsymbol{x}_0 = \boldsymbol{x}$. $d = |\mathcal{X}|$ denotes the input dimension.

**Then we give the proof of Theorem 1.**

*Proof.* According to the Lemma 1, $f$ is the Lipschitz-continuous function, $f_\nu$ has the Lipschitz-continuous gradient. So, according to the property of Lipschitz-continuous gradient,

$$f_{\mu,\nu}(\boldsymbol{x}_{t+1}) \leq f_{\mu,\nu}(\boldsymbol{x}_t) - \eta_t\langle\nabla f_{\mu,\nu}(\boldsymbol{x}_t), g_{\mu,\nu}(\boldsymbol{x}_t)\rangle + \frac{1}{2}\eta_t^2 L_1(f_{\mu,\nu})\|g_{\mu,\nu}(\boldsymbol{x}_t)\|^2 \tag{3}$$

The $g_{\mu,\nu}(\boldsymbol{x}_t)$ can be decomposed into

$$g_{\mu,\nu}(\boldsymbol{x}_t) = g_\mu(\boldsymbol{x}_t + \nu\boldsymbol{v}_{t1}) + \frac{f(\boldsymbol{x}_t + \mu\boldsymbol{u}_t + \nu\boldsymbol{v}_{t1}) - f(\boldsymbol{x}_t + \nu\boldsymbol{v}_{t1}) + f(\boldsymbol{x}_t + \nu\boldsymbol{v}_{t1}) - f(\boldsymbol{x}_t + \nu\boldsymbol{v}_{t2})}{\mu}\boldsymbol{u}_t$$
$$\geq g_\mu(\boldsymbol{x}_t + \nu\boldsymbol{v}_{t1}) - \frac{L_0(f)\nu\|\boldsymbol{v}_{t1} - \boldsymbol{v}_{t2}\|\boldsymbol{u}_t}{\mu} \tag{4}$$

And, following the above decomposition Eq.(4), the last square term of Eq.(3) is bounded by

$$\|g_{\mu,\nu}(\boldsymbol{x}_t)\|^2 \leq \frac{(f(\boldsymbol{x}_t + \mu\boldsymbol{u}_t + \nu\boldsymbol{v}_{t1}) - f(\boldsymbol{x}_t + \nu\boldsymbol{v}_{t1}))^2\|\boldsymbol{u}_t\|^2}{\mu^2} + \frac{(f(\boldsymbol{x}_t + \nu\boldsymbol{v}_{t1}) - f(\boldsymbol{x}_t + \nu\boldsymbol{v}_{t2}))^2\|\boldsymbol{u}_t\|^2}{\mu^2}$$
$$+ \frac{2|f(\boldsymbol{x}_t + \mu\boldsymbol{u}_t + \nu\boldsymbol{v}_{t1}) - f(\boldsymbol{x}_t + \nu\boldsymbol{v}_{t1})| * |f(\boldsymbol{x} + \nu\boldsymbol{v}_{t1}) - f(\boldsymbol{x} + \nu\boldsymbol{v}_{t2})|}{\mu^2}\|\boldsymbol{u}_t\|^2$$
$$\leq \|g_u(\boldsymbol{x} + \nu\boldsymbol{v}_{t1})\|^2 + \frac{L_0(f)^2\nu^2}{\mu^2}\|\boldsymbol{v}_{t1} - \boldsymbol{v}_{t2}\|^2\|\boldsymbol{u}_t\|^2 + 2\frac{L_0(f)^2\nu}{\mu}\|\boldsymbol{v}_{t1} - \boldsymbol{v}_{t2}\|\|\boldsymbol{u}_t\|^3 \tag{5}$$

Then, we take the expectation over $\boldsymbol{u}_t$, $\boldsymbol{v}_{t1}$ and $\boldsymbol{v}_{t2}$. And since $\boldsymbol{v}_{t1}$ and $\boldsymbol{v}_{t2}$ are identically independent with each other, $\boldsymbol{v}_{t1} - \boldsymbol{v}_{t2}$ is still Gaussian random variable. So, according to Lemma 3, we have

$$\mathbb{E}_{u_t,v_{t1,t2}}(f_{\mu,\nu}(\boldsymbol{x}_{t+1})) \leq f_{\mu,\nu}(\boldsymbol{x}_t) - \eta_t\|\nabla f_{\mu,\nu}(\boldsymbol{x}_t)\|^2 + \frac{1}{2}\eta_t^2 L_1(f_{\mu,\nu})(L_0(f)^2(d+4)^2$$
$$+ \frac{2L_0(f)^2\nu^2}{\mu^2}d^2 + \frac{2\sqrt{2}L_0(f)^2\nu}{\mu}(d+3)^{\frac{3}{2}}d^{\frac{1}{2}}) \tag{6}$$

And then use Lemma 1, we have $L_1(f_{\mu,\nu}) \leq L_0(f_\nu) \leq L_0(f)$, so

$$\mathbb{E}_{u_t,v_{t1,t2}}(f_{\mu,\nu}(\boldsymbol{x}_{t+1})) \leq f_{\mu,\nu}(\boldsymbol{x}_t) - \eta_t\|\nabla f_{\mu,\nu}(\boldsymbol{x}_t)\|^2 + \frac{1}{2}\eta_t^2\frac{L_0(f)^3}{\mu}(d+4)^2 d^{\frac{1}{2}}$$
$$+ \eta_t^2\frac{L_0(f)^3\nu^2}{\mu^3}d^{\frac{5}{2}} + \eta_t^2\frac{\sqrt{2}L_0(f)^3\nu}{\mu^2}(d+3)^{\frac{3}{2}}d$$

We take the expectation on $\mathcal{U}_t, \mathcal{V}_t$ and

$$\mathbb{E}_{\mathcal{U}_t, \mathcal{V}_t}(f_{\mu,\nu}(\boldsymbol{x}_{t+1})) \leq \mathbb{E}_{\mathcal{U}_{t-1}, \mathcal{V}_{t-1}}(f_{\mu,\nu}(\boldsymbol{x}_t)) - \eta_t \mathbb{E}_{\mathcal{U}_t, \mathcal{V}_t}(\|\nabla f_{\mu,\nu}(\boldsymbol{x}_t)\|^2) + \frac{1}{2}\eta_t^2 \frac{L_0(f)^3}{\mu}(d+4)^2 d^{\frac{1}{2}}$$

$$+ \eta_t^2 \frac{L_0(f)^3 \nu^2}{\mu^3} d^{\frac{5}{2}} + \eta_t^2 \frac{\sqrt{2}L_0(f)^3 \nu}{\mu^2}(d+3)^{\frac{3}{2}} d$$

For our black-box attacks problem, data dimension is very high ($10^5 \sim 10^7$). So, we have

$$\mathbb{E}_{\mathcal{U}_t, \mathcal{V}_t}(f_{\mu,\nu}(\boldsymbol{x}_{t+1})) \leq \mathbb{E}_{\mathcal{U}_{t-1}, \mathcal{V}_{t-1}}(f_{\mu,\nu}(\boldsymbol{x}_t)) - \eta_t \mathbb{E}_{\mathcal{U}_t, \mathcal{V}_t}(\|\nabla f_{\mu,\nu}(\boldsymbol{x}_t)\|^2)$$

$$+ \eta_t^2 L_0(f)^3 d^{\frac{5}{2}}(\frac{1}{2\mu} + \frac{\sqrt{2}\nu}{\mu^2} + \frac{\nu^2}{\mu^3})$$

Using the same reasoning, we get

$$\mathbb{E}_{\mathcal{U}_0, \mathcal{V}_0}(f_{\mu,\nu}(\boldsymbol{x}_1)) \leq (f_{\mu,\nu}(\boldsymbol{x}_0)) - \eta_t \mathbb{E}_{\mathcal{U}_0, \mathcal{V}_0}(\|\nabla f_{\mu,\nu}(\boldsymbol{x}_0)\|^2)$$

$$+ \eta_t^2 L_0(f)^3 d^{\frac{5}{2}}(\frac{1}{2\mu} + \frac{\sqrt{2}\nu}{\mu^2} + \frac{\nu^2}{\mu^3})$$

Summing up these inequalities, denote $S_Q = \sum_{t=0}^{Q} \eta_t$. And according to the property $f_\mu(\boldsymbol{x}) \geq f(\boldsymbol{x})$, we also have $f_{\mu,\nu}(\boldsymbol{x}) \geq f_\nu(\boldsymbol{x})$. So we get

$$\frac{1}{S_Q}\sum_{t=0}^{Q} \eta_t \mathbb{E}_{\mathcal{U}_t, \mathcal{V}_t}(\|\nabla f_{\mu,\nu}(\boldsymbol{x}_t)\|^2) \leq \frac{1}{S_Q}(f_{\mu,\nu}(\boldsymbol{x}_0) - f_\nu^*) + \frac{1}{S_Q}\sum_{t=0}^{Q} \eta_t^2 L_0(f)^3 d^{\frac{5}{2}}(\frac{1}{2\mu} + \frac{\sqrt{2}\nu}{\mu^2} + \frac{\nu^2}{\mu^3})$$

Here, in order to bound the gap $\epsilon$ between $f_{\mu,\nu}(x)$ and $f_\nu(x)$, we could choose $\mu \leq \hat{\mu} = \frac{\epsilon}{d^{\frac{1}{2}}L_0(f)}$ like without adding noise [21]. So we have

$$\frac{1}{S_Q}\sum_{t=0}^{Q} \eta_t \mathbb{E}_{\mathcal{U}_t, \mathcal{V}_t}(\|\nabla f_{\mu,\nu}(\boldsymbol{x}_t)\|^2) \leq \frac{1}{S_Q}(f_{\mu,\nu}(\boldsymbol{x}_0) - f_\nu^*) + \frac{1}{S_Q}\sum_{t=0}^{Q} \frac{\eta_t^2 L_0(f)^4 d^3}{\epsilon}(\frac{1}{2} + \frac{\sqrt{2}\nu}{\mu} + \frac{\nu^2}{\mu^2})$$

We take the constant stepsize and set $\eta_t = \eta$. We also denote $\alpha = \frac{\nu}{\mu}$. And we set $(\frac{1}{2} + \sqrt{2}\alpha + \alpha^2)$ as $\gamma(\alpha)$ which is increasing function of the ratio $\frac{\nu}{\mu}$. We minimize the right hand side then get

$$\eta = \left[\frac{R\epsilon}{d^3 L_0^3(f)(Q+1)}\right]^{1/2} \frac{1}{\sqrt{\gamma(\alpha)}}$$

So we can get

$$\frac{1}{Q+1}\sum_{t=0}^{Q} \mathbb{E}_{\mathcal{U}_{t-1}, \mathcal{V}_{t-1}}(\|\nabla f_{\mu,\nu}(\boldsymbol{x}_t)\|^2) \leq \frac{2L_0(f)^{\frac{5}{2}} R^{\frac{1}{2}} d^{\frac{3}{2}}}{(Q+1)^{\frac{1}{2}} \epsilon^{\frac{1}{2}}}\sqrt{\gamma(\alpha)} \tag{7}$$

In order to guarantee the expected squared norm of the gradient of function $f_{\mu,\nu}$ of the order $\delta$, the lower bound for the expected number of queries is

$$O\left(\gamma(\alpha)\frac{d^3 L_0^5(f)R}{\epsilon\delta^2}\right)$$

$\square$

# E   Proof of Section 4.2.2

## E.1   The Non-Convex and Smooth Case

Now, the gradient estimator in ZO attacks becomes

$$\tilde{g}_{\mu,\nu}(\boldsymbol{x}) = \frac{1}{M}\sum_{j=1}^{M} \frac{f(\boldsymbol{x} + \mu\boldsymbol{u} + \nu\boldsymbol{v}_{j1}) - f(\boldsymbol{x} + \nu\boldsymbol{v}_{j2})}{\mu}\boldsymbol{\mu} \tag{8}$$

We denote the sequence of standard Gaussian noises added by the attacker as $\mathcal{U}_t = \{\boldsymbol{u}_0, \boldsymbol{u}_1, \ldots, \boldsymbol{u}_t\}$. Note that here the definition of the sequential standard Gaussian noises added by the defende should be updated to $\mathcal{V}_t = \{\boldsymbol{v}_{01}, \ldots, \boldsymbol{v}_{0M}, \ldots, \boldsymbol{v}_{t1}, \ldots, \boldsymbol{v}_{tM}\}$. $\boldsymbol{v}_{ij} \in \mathcal{V}_t$ contains $\boldsymbol{v}_{ij1}$ and $\boldsymbol{v}_{ij2}$.

**Then we give the proof of Theorem 2.**

*Proof.* Followed by proof of last section, according to the property of Lipschitz-continuous gradient,

$$f_{\mu,\nu}(\boldsymbol{x}_{t+1}) \leq f_{\mu,\nu}(\boldsymbol{x}_t) - \eta_t \langle \nabla f_{\mu,\nu}(\boldsymbol{x}_t), \tilde{g}_{\mu,\nu}(\boldsymbol{x}_t) \rangle + \frac{1}{2}\eta_t^2 L_1(f_{\mu,\nu}) \|\tilde{g}_{\mu,\nu}(\boldsymbol{x}_t)\|^2 \quad (9)$$

Followed by Eq.(4), the $\tilde{g}_{\mu,\nu}(\boldsymbol{x}_t)$ can be also decomposed into

$$\tilde{g}_{\mu,\nu}(\boldsymbol{x}_t) \geq \sum_{j=1}^{M} \frac{1}{M} g_\mu(\boldsymbol{x}_t + \nu \boldsymbol{v}_{t1j}) - \frac{L_0(f)\nu \|\boldsymbol{v}_{t1} - \boldsymbol{v}_{t2}\| \boldsymbol{u}_t}{\mu} \quad (10)$$

Then, to bound the square term of gradient estimator, we use the decomposition

$$\|\tilde{g}_{\mu,\nu}(\boldsymbol{x}_t)\|^2 = \|g_\mu(\boldsymbol{x}_t)\|^2 + \|(\tilde{g}_{\mu,\nu}(\boldsymbol{x}_t) - g_\mu(\boldsymbol{x}_t))\|^2 + 2\langle g_\mu(\boldsymbol{x}_t), (\tilde{g}_{\mu,\nu}(\boldsymbol{x}_t) - g_\mu(\boldsymbol{x}_t)) \rangle \quad (11)$$

So, we have

$$\begin{aligned}
\mathbb{E}_{\boldsymbol{v}_{t1-tM}, \boldsymbol{u}_t}(f_{\mu,\nu}(\boldsymbol{x}_{t+1})) \leq & f_{\mu,\nu}(\boldsymbol{x}_t) - \eta_t \mathbb{E}_{\boldsymbol{v}_{t1-tM}, \boldsymbol{u}_t}(\langle \nabla f_{\mu,\nu}(\boldsymbol{x}_t), \tilde{g}_{\mu,\nu}(\boldsymbol{x}_t) \rangle) \\
& + \frac{1}{2}\eta_t^2 L_1(f_{\mu,\nu}) \mathbb{E}_{\boldsymbol{v}_{t1-tM}, \boldsymbol{u}_t}(\|\tilde{g}_{\mu,\nu}(\boldsymbol{x}_t)\|^2) \\
\leq & f_{\mu,\nu}(\boldsymbol{x}_t) - \eta_t \mathbb{E}_{\boldsymbol{v}_{t1-tM}, \boldsymbol{u}_t}(\langle \nabla f_{\mu,\nu}(\boldsymbol{x}_t), \sum_{j=1}^{M} \frac{1}{M} g_\mu(\boldsymbol{x}_t + \nu \boldsymbol{v}_{t1j}) \rangle) \\
& + \frac{1}{2}\eta_t^2 L_1(f_{\mu,\nu}) \mathbb{E}_{\boldsymbol{v}_{t1-tM}, \boldsymbol{u}_t}(\|g_\mu(\boldsymbol{x}_t)\|^2 + \|(\tilde{g}_{\mu,\nu}(\boldsymbol{x}_t) - g_\mu(\boldsymbol{x}_t))\|^2 \\
& + 2\langle g_\mu(\boldsymbol{x}_t), (\tilde{g}_{\mu,\nu}(\boldsymbol{x}_t) - g_\mu(\boldsymbol{x}_t)) \rangle)
\end{aligned} \quad (12)$$

Since $\boldsymbol{v}_{t1-tM}$ are iid random variables, the second term of above inequality is same as the second term of Eq.(6). Then we need to bound the last square terms.

We set the expectation over $\boldsymbol{v}_{1j}$, $\mathbb{E}_{\boldsymbol{v}_{1j}}(f(\boldsymbol{x}+\mu\boldsymbol{u})+\nu\boldsymbol{v}_{1j})$ as $E_1$ and $\mathbb{E}_{\boldsymbol{v}_{2j}}(f(\boldsymbol{x}+\nu\boldsymbol{v}_{2j})$ as $E_2$. And we also set $B_1 = E_1 - f(\boldsymbol{x}+\mu\boldsymbol{u})$ and $B_2 = E_2 - f(\boldsymbol{x})$. We also set $\sigma_1^2 = Var_{\boldsymbol{v}_{1j}}(f(\boldsymbol{x}+\mu\boldsymbol{u}+\nu\boldsymbol{v}_{1j}))$ and $\sigma_2^2 = Var_{\boldsymbol{v}_{2j}}(f(\boldsymbol{x}+\nu\boldsymbol{v}_{2j}))$. Then we bound the second term of above square terms.

$$\begin{aligned}
\mathbb{E}_{\boldsymbol{v}_{t1-tM}}(\|(\tilde{g}_{\mu,\nu}(\boldsymbol{x}_t) - g_\mu(\boldsymbol{x}_t))\|^2) \leq & (B_1^2 + B_2^2 + \frac{\sigma_1^2 + \sigma_2^2}{M} - 2B_1 B_2)\frac{1}{\mu^2}\|\boldsymbol{u}\|^2 \\
\leq & (2(B_1^2 + B_2^2) + \frac{\sigma_1^2 + \sigma_2^2}{M})\frac{1}{\mu^2}\|\boldsymbol{u}\|^2
\end{aligned} \quad (13)$$

We can compute this square of sum directly and we need to bound $B^2$ and $\sigma^2$. By using the Taylor expansion of $f(\boldsymbol{x})$,

$$\begin{aligned}
f(\boldsymbol{x}+\mu\boldsymbol{u}+\nu\boldsymbol{v}_{1j}) \leq & f(\boldsymbol{x}+\mu\boldsymbol{u}) + \nabla f(\boldsymbol{x}+\mu\boldsymbol{u})^T \nu\boldsymbol{v}_{1j} + \frac{1}{2}\nu^2 L_1(f)\|\boldsymbol{v}_{1j}\|^2 \\
E_1 \leq & f(\boldsymbol{x}+\mu\boldsymbol{u}) + \frac{1}{2}\nu^2 L_1(f)d \\
B_1 = & E_1 - f(\boldsymbol{x}+\mu\boldsymbol{u}) \leq \frac{1}{2}\nu^2 L_1(f)d \\
B_1 = & E_1 - f(\boldsymbol{x}+\mu\boldsymbol{u}) \geq -\frac{1}{2}\nu^2 L_1(f)d
\end{aligned}$$

So, we have

$$f(\boldsymbol{x}+\mu\boldsymbol{u}+\nu\boldsymbol{v}_{1j}) - f(\boldsymbol{x}+\mu\boldsymbol{u}) - \frac{1}{2}\nu^2 L_1(f)d \leq f(\boldsymbol{x}+\mu\boldsymbol{u}+\nu\boldsymbol{v}_{1j}) - E_1 \leq f(\boldsymbol{x}+\mu\boldsymbol{u}+\nu\boldsymbol{v}_{1j}) - f(\boldsymbol{x}+\mu\boldsymbol{u}) + \frac{1}{2}\nu^2 L_1(f)d$$

Based on the above inequality, we have

$$\mathbb{E}_{\boldsymbol{v}_{1j}}((f(\boldsymbol{x}+\mu\boldsymbol{u}+\nu\boldsymbol{v}_{1j})-E_1)^2) \leq 2\mathbb{E}_{\boldsymbol{v}_{1j}}((f(\boldsymbol{x}+\mu\boldsymbol{u}+\nu\boldsymbol{v}_{1j})-f(\boldsymbol{x}+\mu\boldsymbol{u}))^2) + \frac{1}{2}\nu^4 L_1(f)^2 d^2$$

$$\sigma_1^2 \leq 2\nu^2 L_0(f)^2 d + \frac{1}{2}\nu^4 L_1(f)^2 d^2$$

Then, the Eq.(13) becomes

$$\mathbb{E}_{\boldsymbol{v}_{t1-tM},\boldsymbol{u}}(\|(\tilde{g}_{\mu,\nu}(\boldsymbol{x}_t)-g_\mu(\boldsymbol{x}_t))\|^2) \leq \mathbb{E}_{\boldsymbol{u}}((B_1^2 + B_2^2 + \frac{\sigma_1^2+\sigma_2^2}{M} - 2B_1 B_2)\frac{1}{\mu^2}\|\boldsymbol{u}\|^2)$$

$$\leq \mathbb{E}_{\boldsymbol{u}}((2(B_1^2+B_2^2) + \frac{\sigma_1^2+\sigma_2^2}{M})\frac{1}{\mu^2}\|\boldsymbol{u}\|^2) \qquad (14)$$

$$\leq \frac{\nu^4}{\mu^2}L_1(f)^2 d^3 + \frac{\nu^4}{\mu^2 M}L_1(f)^2 d^3 + 4\frac{\nu^2}{\mu^2 M}L_0(f)^2 d^2$$

And the third term of square terms can be bounded by using $B$ and $E$.

$$2(f(\boldsymbol{x}+\mu u)-f(\boldsymbol{x}))(\sum_{j=1}^M(\frac{f(\boldsymbol{x}+\mu\boldsymbol{u}+\nu\boldsymbol{v}_{1j})-f(\boldsymbol{x}+\nu\boldsymbol{v}_{2j})}{M}) - f(\boldsymbol{x}+\mu\boldsymbol{u}) + f(\boldsymbol{x}))\frac{1}{\mu^2}\|\boldsymbol{u}\|^2$$

$$\leq 2(f(\boldsymbol{x}+\mu\boldsymbol{u})-f(\boldsymbol{x}))(E_1-E_2-f(\boldsymbol{x}+\mu\boldsymbol{u})+f(\boldsymbol{x}))\frac{1}{\mu^2}\|\boldsymbol{u}\|^2 \quad \text{\#taking expectation over } \boldsymbol{v}_{t1-tM},\boldsymbol{v}_{2j}$$

$$\leq 2\frac{L_0(f)\mu\|\boldsymbol{u}\|(B_1-B_2)\|\boldsymbol{u}\|^2}{\mu^2} \quad \text{\#taking expectation over } \boldsymbol{u}$$

$$\leq 2\frac{\nu^2}{\mu}L_0(f)L_1(f)d^{\frac{5}{2}}$$

$$(15)$$

So, we have

$$\mathbb{E}_{\boldsymbol{v}_{t1-tM},\boldsymbol{u}_t}\|\tilde{g}_{\mu,\nu}(\boldsymbol{x}_t)\|^2 \leq L_0(f)^2(d+4)^2 + 4\frac{\nu^2 L_0(f)^2}{\mu^2 M}d^2 + 2\frac{\nu^2 L_0(f)L_1(f)}{\mu}d^{\frac{5}{2}}$$

$$+ \frac{\nu^4 L_1(f)^2}{\mu^2}\frac{M+1}{M}d^3$$

And based on Lemma 2, the Eq.(21) becomes

$$\mathbb{E}_{\boldsymbol{v}_{t1-tM},\boldsymbol{u}_t}(f_{\mu,\nu}(\boldsymbol{x}_{t+1})) \leq f_{\mu,\nu}(\boldsymbol{x}_t) - \eta_t\|\nabla f_{\mu,\nu}(\boldsymbol{x}_t)\|^2 + \frac{1}{2}\eta_t^2 L_1(f_{\mu,\nu})\mathbb{E}_{\boldsymbol{v}_{t1-tM},\boldsymbol{u}_t}(\|\tilde{g}_{\mu,\nu}(\boldsymbol{x}_t)\|^2)$$

$$\leq f_{\mu,\nu}(\boldsymbol{x}_t) - \eta_t\|\nabla f_{\mu,\nu}(\boldsymbol{x}_t)\|^2 + \eta_t^2(\frac{L_0(f)^2 L_1(f)}{2}d^2$$

$$+ \frac{2\nu^2 L_0(f)^2 L_1(f)}{\mu^2 M}d^2 + \frac{\nu^2 L_0(f)L_1(f)^2}{\mu}d^{\frac{5}{2}} + \frac{\nu^4 L_1(f)^3(M+1)}{2\mu^2 M}d^3)$$

$$(16)$$

We take the expectation on $\mathcal{U}_t$, $\mathcal{V}_t$,

$$\mathbb{E}_{\mathcal{U}_t,\mathcal{V}_t}(f_{\mu,\nu}(\boldsymbol{x}_{t+1})) \leq \mathbb{E}_{\mathcal{U}_{t-1},\mathcal{V}_{t-1}}(f_{\mu,\nu}(\boldsymbol{x}_t)) - \eta_t\mathbb{E}_{\mathcal{U}_t,\mathcal{V}_t}(\|\nabla f_{\mu,\nu}(\boldsymbol{x}_t)\|^2) + \eta_t^2(\frac{L_0(f)^2 L_1(f)}{2}d^2$$

$$+ \frac{2\nu^2 L_0(f)^2 L_1(f)}{\mu^2 M}d^2 + \frac{\nu^2 L_0(f)L_1(f)^2}{\mu}d^{\frac{5}{2}} + \frac{\nu^4 L_1(f)^3(M+1)}{2\mu^2 M}d^3)$$

$$(17)$$

Then we can get

$$\frac{1}{S_Q}\sum_{t=0}^{Q}\eta_t\mathbb{E}_{\mathcal{U}_t,\mathcal{V}_t}(\|\nabla f_{\mu,\nu}(\boldsymbol{x}_t)\|^2) \leq \frac{1}{S_Q}(f_{\mu,\nu}(\boldsymbol{x}_0) - f_{\nu}^*) + \frac{1}{S_Q}\sum_{t=0}^{Q}\eta_t^2(\frac{L_0(f)^2 L_1(f)}{2}d^2$$
$$+ \frac{2\nu^2 L_0(f)^2 L_1(f)}{\mu^2 M}d^2 + \frac{\nu^2 L_0(f)L_1(f)^2}{\mu}d^{\frac{5}{2}} \qquad (18)$$
$$+ \frac{\nu^4 L_1(f)^3(M+1)}{2\mu^2 M}d^3)$$

$\square$

# F    The Proof of Section 4.3

The direction searching of search-based attacks [1, 2, 8, 13, 20] can be formulated as

$$s(\boldsymbol{x}) = \mathbb{I}(f(\boldsymbol{x} + \mu\boldsymbol{u}) - f(\boldsymbol{x}) < 0) \cdot \mu\boldsymbol{u}$$
$$= \mathbb{I}(h(\boldsymbol{x}) < 0) \cdot \mu\boldsymbol{u} \qquad (19)$$

where $\boldsymbol{u}$ is the direction searching direction sampled from some pre-defined distributions, such as gaussian noise in [8], orthogonal basis in [13] and squared perturbations in [2].

If the attackers take the attack direction $\boldsymbol{u}$ and objective function decreases, then $\boldsymbol{u}$ will be seen as the potential attack direction.

Now, with the RND, the searched direction becomes

$$s_\nu(\boldsymbol{x}) = \mathbb{I}(f(\boldsymbol{x} + \mu\boldsymbol{u} + \nu\boldsymbol{v}_1) - f(\boldsymbol{x} + \nu\boldsymbol{v}_2) < 0) \cdot \mu\boldsymbol{u}$$
$$= \mathbb{I}(h_\nu(\boldsymbol{x}) < 0) \cdot \mu\boldsymbol{u} \qquad (20)$$

**We give the Theorem 3 about the probability of $\mathbf{Sign}(h(\boldsymbol{x})) \neq \mathbf{Sign}(h_\nu(\boldsymbol{x}))$,**

$$P(\mathrm{Sign}(h(\boldsymbol{x})) \neq \mathrm{Sign}(h_\nu(\boldsymbol{x})) \leq P(h_\nu(\boldsymbol{x}) - h(\boldsymbol{x})| \geq |h(\boldsymbol{x})|)$$
$$\leq \frac{\mathbb{E}[|h_\nu(\boldsymbol{x}) - h(\boldsymbol{x})|]}{|h(\boldsymbol{x})|} \text{ according to the Markov's inequality}$$
$$\leq \frac{\sqrt{\mathbb{E}[(h_\nu(\boldsymbol{x}) - h(\boldsymbol{x}))^2]}}{|h(\boldsymbol{x})|} \text{ according to the Jensen's inequality}$$
$$\leq \frac{\sqrt{\mathbb{E}[2(f(\boldsymbol{x} + \mu\boldsymbol{u} + \nu\boldsymbol{v}_1) - f(\boldsymbol{x} + \mu\boldsymbol{u}))^2 + 2(f(\boldsymbol{x} + \nu\boldsymbol{v}_2) - f(\boldsymbol{x}))^2]}}{|h(\boldsymbol{x})|}$$
$$\leq \frac{\sqrt{\mathbb{E}[2L_0(f)^2\nu^2\|\boldsymbol{v}_1\|^2 + 2L_0(f)^2\nu^2\|\boldsymbol{v}_2\|^2]}}{|h(\boldsymbol{x})|} \text{ according to Lipchitzness of function}$$
$$\leq \frac{2L_0(f)\nu\sqrt{d}}{|h(\boldsymbol{x})|} \text{ according to Lemma 3}$$

$$(21)$$