# OpenReview forum: "Random Noise Defense Against Query-Based Black-Box Attacks"
_NeurIPS.cc/2021/Conference — NeurIPS 2021 Poster_

### Official Review · Reviewer_Q2xR · 2021-07-16

**Rating:** 6
**Confidence:** 3

**Summary:**

This work proposes addition of Gaussian noise as a defense against black box attacks and analyses the convergence of query attacks with this defense enabled. The work is quite sound.


**Limitations And Societal Impact:**

Adequately addressed

**Main Review:**

I have few questions wrt experiments


- What’s the motivation to choose a different \nu for RND, RND-GF and RND-AT in Table 2?
- RND by itself seemed quite poor against NES and Square attack while RND-GF provides quite a boost. Any insights on why this is the case?
- There is a significant drop in clean accuracy with RND-AT, any suggestions on what hyperparameters could help tune this better?
- What would you prescribe as a method to choose noise hyperparameters?
- On a more general note, does the noise need to be Gaussian, or is there an intuition on the suitability of other quantisation noises?
- Tables with small font size were a bit challenging to parse

**Time Spent Reviewing:**

3

---

> ### Author Response · Authors · 2021-08-10
> **Thank you for your comments and recognition of our work!**
>
> ### Q1. "What’s the motivation to choose a different \nu for RND, RND-GF and RND-AT in Table 2?"
>
>
> In Table 2, we set $\nu$ as 0.02, 0.05, and 0.05 for RND, RND-GF, and RND-AT, respectively. According to our theoretical analysis of RND, the larger $\nu$ can increase the defense performance but may cause more significant harm to the clean accuracy. As shown in Figure 1 (d), for clean model, RND with large $\nu$ ($\nu >0.02$) can largely affect clean accuracy. For GF and AT models, they are less sensitive to small random noise. Therefore, we adopt $\nu=0.02$ for RND and adopt the larger $\nu$ ($\nu=0.05$) for RND-GF and RND-AT, while maintaining a relatively high clean accuracy. We presented some brief illustrations in Line 348-350 and Line 358-360.
>
> And we will clarify the motivation of this setting in the revised version.
>
> ---
>
> ### Q2. "RND by itself seemed quite poor against NES and Square attack while RND-GF provides quite a boost. Any insights on why this is the case?"
>
> Thanks for this insightful comment.
>
> **1)** As shown in Table 2, to maintain the high clean accuracy, RND is realized by adding a small Gaussian noise (i.e., $\nu = 0.02$) to each query. Compared to the clean model, RND can consistently improve the defense performance. The attack failure rate of Square is increased from 0.02 to 0.116 on CIFAR-10 and from 0.0 to 0.101 on ImageNet, and the attack failure rate of NES is increased from 0.01 to 0.05 on CIFAR-10 and from 0.0 to 0.245 on ImageNet. However, such defense performance is relatively poor compared to other defense methods, including RND-GF.
>
> **2)** In contrast, the model with GF allows a relative larger value of $\nu$, to ensure that the clean accuracy will not significantly decrease (as sown in Figure 1(d)). In Table 2, we adopted $\nu=0.05$ for RND-GF. According to our theoretical analysis, larger $\nu$ will lead to better defense performance.
>
> We will clarify the above points in the revised version.
>
>
> ---
>
> ### Q3. "There is a significant drop in clean accuracy with RND-AT, any suggestions on what hyperparameters could help tune this better?"
>
>
> The drop in clean accuracy of RND-AT comes in two aspects:
> **1)** adversarial training. For AT models in our experiments, we adopt the pretrained AT model in [1] on CIFAR-10 and pretrained AT model in Robustness Library [2] on ImageNet. For the adopted AT models, the clean accuracy decreases from 96.60% to 89.48% on CIFAR-10 and from 74.90% to 61.60% on ImageNet, and **2)** by adding random noise to each query, the RND-AT further decreases the clean accuracy of AT from 89.48% to 87.40% on CIFAR-10 and from 61.60% to 58.15% on ImageNet.
>
> The performance drop is mainly caused by the trade-off between robustness and clean accuracy in AT, and one can tune the trade-off parameters in AT towards better clean performance with a decline of the adversarial robustness. For the performance drop caused by RND, it is suggested to perform Gaussian fine-tuning to better maintain the clean accuracy.
>
>
> [1] Uncovering the Limits of Adversarial Training against Norm-Bounded Adversarial Examples, Gowal et al., arxiv 2021
>
> [2] Robustness (Python Library), Engstrom et al., 2019
>
> ---
>
> ### Q4. "What would you prescribe as a method to choose noise hyperparameters?"
>
> Based on our theoretical analysis, the larger $\nu$ leads to the better defense performance. However, the larger $\nu$ will also affect the clean accuracy more significantly. In practice, to obtain a suitable value of $\nu$, one can evaluate the clean accuracy on the validation set with different values of $\nu$, and select the largest $\nu$ with acceptable clean performance. Also, as verfied in our experiments, the Gaussian fine-tuning can also boost the clean performane of RND, to enable a relative larger value of $\nu$.
>
> ---
>
> ### Q5. "On a more general note, does the noise need to be Gaussian, or is there an intuition on the suitability of other quantisation noises?"
>
> Thanks for your constructive comment! We believe that other forms of noise could also be adopted, such as noises sampled from Uniform distribution or Bernoulli distribution (for each pixel). For example, in Theorem 3 of Sec. 4.3, the probability P of different sign between $h(x)$ and $h_{\nu}(x)$ is controlled by the relative values of the $h_{\nu}(x)$ and $h(x)$. The higher P, the attackers are more likely to be misled and need more queries to find adversarial examples. And, given the value $\mu$ for attackers and the local linearity of model, the relative values $h(x)$ and $h_{\nu}(x)$ are controlled by the size of defense noise. So, they still follow the statement that the larger defense noise could lead to better defense performance. We will further analyze and evaluate the other noise forms in our future work.
>
> ---
>
> ### Q6. "Tables with small font size were a bit challenging to parse"
>
> Thanks for this suggestion!  We will tune the font size of the tables in our revised version.

---

### Official Review · Reviewer_CW2N · 2021-07-17

**Rating:** 6
**Confidence:** 4

**Summary:**

This paper proposes to use random Gaussian noise as defense for black-box attacks. Also, a Gaussian augmentation Fine-tuning is also proposed to enable adding larger noise.


**Limitations And Societal Impact:**

Please see my main review

**Main Review:**

Overall, this paper is well motivated. The analysis is straight and well-understood.

In table 2, can you explain why RND-GF is even better than GF in terms of clean acc?

**Time Spent Reviewing:**

4h

---

> ### Author Response · Authors · 2021-08-10
> **Thanks for your comments and recognition of our work!**
>
> ### Q1. "In table 2, can you explain why RND-GF is even better than GF in terms of clean acc?
>
> Thanks for your careful observation.
>
> For fine-tuning the model with Gaussian augmentation (GF), the distribution of the training set is denoted as $P(X + Gaussian)$. If the fine-tuned model fits this distribution very well, then it is possible that the accuracy on the testing data sampled from $P(X + Gaussian)$ (just like testing with RND) is higher than the accucary on testing data sampled from $P(X)$ (just like testing on clean samples without RND). This is the possible reason that the clean accuracy of RND-GF (i.e., $92.4\%$) is higher than that of GF (i.e., $91.72\%$) on CIFAR-10, as shown in Table 2.
>
> In contrast, on ImageNet, the clean accuracy of RND-GF (i.e., $71.15\%$) is lower than that of GF (i.e., $74.7\%$). It demonstrates that the GF model doesn't fit $P(X + Gaussian)$ on ImageNet very well, compared to CIFAR-10. Actually, it is reasonable, because the Gaussian noise used in fine-tuning process is $N(0; 0.5I)$ on ImageNet, while $N(0; 0.1I)$ on CIFAR-10. Moreover, the input dimensionality of ImageNet is much higher than that of CIFAR-10. Larger Gaussian noise and higher input dimensionality could largely increase the training difficulty and may require more epoches to achieve better fit of $P(X+Gaussian)$. However, in our experiments, the fine-tuning epoches of both ImageNet and CIFAR-10 are set to 50. We will add more thorough emperical studies to analyze the effect of GF and RND on the clean accuracy.

---

### Official Review · Reviewer_opkE · 2021-07-17

**Rating:** 7
**Confidence:** 3

**Summary:**

This paper proposes a simple but theoretically supported defense method against query-based black-box attack, called as Random Noise Defense. The core idea is to add Gaussian noise to a given input image at each query so it can be considered as quite simple. However, this paper shows theoretical analyses of RND against zero-order optimization, adaptive and search-based attacks. The overall finding through each theoretical analysis is that the number of queries for successful attacks by an attacker changes with the ratio of nu/mu. To overcome trade-off between clean accuracy and larger nu, the authors propose Gaussian augmentation fine-tuning for the target model. From the extensive experiments using CIFAR-10 and ImageNet against various black-box attacks, the authors demonstrated that, not only the results of theoretical analysis are confirmed by experiments but RND achieves the superior performance. Also, they showed RND can be easily combined with other defense methods and it achieves the state-of-the-art defense performance when it is combined with the adversarially-trained model.

**Ethical Concerns:**

Ethical Concerns

There is no ethical concerns in this paper.

**Limitations And Societal Impact:**

Limitations

1. In my opinion, Assumption 1 is somewhat acceptable for a neural net-based classifier. However, I wonder if Assumption 2 is also generally accepted because it stands for a higher level of smooth functions.

2. Input transformation-based defense methods (e.g. R&P, JPEG and bit-red) are should be considered as a baseline because [1] reported that these methods relatively achieve the superior robustness than other types of defense in case of query-based black-box attacks.

3. I think that there is a methodical connection or similarity between the proposed method and randomized smoothing-based methods [13, 38, 39]. However, this paper seems to lack reference to that content. Therefore, it would be a more solid paper if the authors address differences in theoretical and methodological perspectives from existing similar methods.


[1] Benchmarking Adversarial Robustness on Image Classification, Y. Dong et. al., CVPR 2020


Societal Impact

There is no potential negative social impact in this paper.

**Main Review:**

Originality

The proposed idea, adding Gaussian noise to an input image, has been used for adversarial defense, such as randomized smoothing.
However, This paper has the following differences from it. Firstly, the proposed method is focused on adversarial defense against query-based black-box attack. Second, from a different perspective, this paper theoretically and experimentally shows the superiority of the proposed method. Therefore, this paper has its own contribution.

Quality

I didn't find any flaws in theoretical analysis, but I'm not sure because I am not a theoretical expert.
The authors demonstrated the strength and theoretical findings of the proposed method from the extensive and solid experiments well.
Also, they stated limitations and future works of their work.

Clarity

This paper is well written and easy to read.

Significance

The results of this paper is important because this paper proposes a simple defense method as well as a theoretical analysis of it.


**Time Spent Reviewing:**

6

---

> ### Author Response · Authors · 2021-08-10
> **Thank you for your comments and recognition of our work!**
>
> ### Q1. "In my opinion, Assumption 1 is somewhat ...... higher level of smooth functions."
>
> Thanks for this insightful comment.
>
> **1)** Assumption 2 is only used in Theorem 2. As shown in the proof of Theorem 2 (see Line 178 and Eq. (9) in the supplementary pdf), Assumption 2 (i.e., the property of Lipschitz-continuous gradient) is applied to capture the relationship between $f_{\mu, \nu}(x_{t+1})$ and $f_{\mu, \nu}(x_{t})$. In the scenario of iterative attack, the perturbation size of each step is often very small, thus $x_{t+1}$ is close to $x_{t}$. It means that we only require that Assumption 2 holds in the very small local region. That is a practical requirement, and the alignment between the experimental evaluations and Theorem 2 also supports this point.
>
> **2)** we notice that Assumption 2 has been widely used in many existing adversarial attack and defense methods, such as [1][2][3][4][5].
>
> Hope that the above demonstrations can somewhat alleviate the reviewer's concern on Assumption 2. And, we will clarify it more clearly in our revised version, to help future readers better understand the role and practical rationality of Assumption 2.
>
>
> [1] On the Convergence and Robustness of Adversarial Training, Wang et al., ICML 2019
>
>
>
> [2] On the Loss Landscape of Adversarial Training: Identifying Challenges and How to Overcome Them, Liu et al., NeurIPS 2020
>
> [3] signSGD via Zeroth-Order Oracle, Liu et al., ICLR 2019
>
> [4] Sign-OPT: A Query-Efficient Hard-label Adversarial Attack, Cheng et al., ICLR 2020
>
> [5] Square Attack: a query-efficient black-box adversarial attack via random search, Andriushchenko, et al., ECCV 2020
>
>
> ---
>
> ### Q2. "Input transformation-based defense methods (e.g. R&P, JPEG, and bit-red) ...... of defense in case of query-based black-box attacks."
>
> Thanks for this constructive suggestion. We did a quick comparison to R&P, Bit-Red, and JPEG on ImageNet and Inception v3 model, the perturbation budget is 12/255 and the maximal query number is 10000. For RND, we set the $\nu$ as 0.02. The results w.r.t. the standard attack and the adaptive EOT attack with M=5 are given below.
>
> #### $\ell_{\infty}$ Attack result (average number of success attacks/attack failure rate)
> | defense methods|clean accuracy| NES ($\ell_{\infty}$)| SignHunter ($\ell_{\infty}$)| Square ($\ell_{\infty}$)|
> |:---------------:|:---------------:|:---------------:|:---------------:|:---------------:|
> |R&P|74.6%|1368.2/0.378| 345.1/0.223|285.8/0.153|
> |Bit-Red|62.5%|1548.6/0.092|145.4/0.048|356.2/0.021|
> |JPEG|74.2%|1417.4/0.205|156.8/0.146|215.3/0.033|
> |RND|76.6%|2143.4/0.413|173.4/0.532| 251.7/0.207|
> #### $\ell_{\infty}$ EOT Attack result (M = 5 in EOT)
> | defense methods| clean accuracy| NES ($\ell_{\infty}$)| SignHunter ($\ell_{\infty}$)| Square ($\ell_{\infty}$)|
> |:---------------:|:---------------:|:---------------:|:---------------:|:---------------:|
> |R&P|74.6%|2547.6/0.302| 545.6/0.148|656.2/0.069|
> |Bit-Red|62.5%|2682.3/0.031|612.3/0.010|689.3/0.004|
> |JPEG|74.2%|2863.2/0.117|485.6/0.094|712.5/0.009|
> |RND|76.6%| 3240.2/0.365|336.5/0.456|910.9/0.121|
>
> Compared to these input transformation-based defense methods, RND achieves better results in both clean accuracy and defense performance. A thorough comparison will be added in our revision.
>
> ---
>
> ### Q3. “I think that there is a methodical connection or similarity between ...... methodological perspectives from existing similar methods.”
>
> Thanks for this constructive comment. We have presented a brief discussion about random-smoothing (RS) methods, from Line 107-112. Here we try to further highlight the difference between RS and our RND method. Actually, both RS and our method belong to the randomization-based defense methods. However, they have quite different goals and methodologies.
>
> **1)**  RS aims to obtain the certified robustness under the $\ell_2$ norm. For each query x, RS first generates $k$ noise-corrupted copies of x and aggregates the prediction of these $k$ queries (e.g., $k$=100), which places a huge burden on inference time ($k$ times of standard inference) and is not applicable to our lightweight defense scenario.
>
> **2)** In this work, we focus on lightweight defense against query-based black-box attacks. To this goal, RND  perturbs each query only once, without any extra burden on inference time, and our theoretical study focuses on the query complexities, rather than the certified radius considered in RS.
>
> We will clarify the differences between RS and our RND more clearly in the revised version.

---

> > ### Comment · Reviewer_opkE · 2021-09-02
> > **Thanks for the reply**
> >
> > I thank the author's reply to my review. I carefully read other reviews and most of the concerns I had were solved through author’s reply.
> > Therefore, I would like to keep my initial rating.

---

> > > ### Author Response · Authors · 2021-09-02
> > > **Greatly appreciated**
> > >
> > > Dear Reviewer opkE,
> > >
> > > Thanks again for your encouraging and constructive comments, and they are very helpful for improving our work.
> > >
> > > Sincerely,
> > >
> > > Authors

---

### Official Review · Reviewer_RFP8 · 2021-07-19

**Rating:** 5
**Confidence:** 3

**Summary:**

The authors suggest the use of Random Noise Defense (RND) to protect deep learning models from black-box query-based attacks. The authors motivate their proposal via both theoretical and experimental validation -- where RND is a lightweight defense that augments random noise to input queries. The paper further proposes RND-GF that is trained to be robust (smoothened) to random noise -- in order to further improve the adversarial robustness.

**Limitations And Societal Impact:**

Yes

**Main Review:**

### Merits
1. The paper systematically studies decision and score based adversarial attacks, their theoretical implications, and empirical performances
2. RND-GF helps combine adversarial and noise robustness
3. This is one of the first works that studies the relation of random noise to black-box query complexity theoretically.

### Qualms
1. **Adaptive nature of attacks:** In adaptive attacks, EOT is the only attack considered. In my opinion, the knowledge of v and mu (only the magnitudes and not the exact noise) is one of the minimum knowledge that should be provided to any adaptive attacker. Moreover, all the theorems are heavily dependent on the ratio of v/mu. Can you elaborate in greater detail, the motivation behind this choice -- and how do the results fair when the knowledge is available to the attacker.

2. **Scope of work:** The scope of the work, though important, is limited. It does not even consider all the types of black-box query-based attacks -- such as transfer adversaries; and is not applicable in the case of non-query attacks.

3. **Failure Rates:** From the empirical failure rates, it can be seen that at the optimal selection of mu by the attacker, the failure rates can be very high. Given this information (in an empirical setting) -- would you suggest that RND is still beneficial to model robustness? (Given that adding random noise has a small tradeoff on clean accuracy as well)

**Time Spent Reviewing:**

6

---

> ### Author Response · Authors · 2021-08-10
> **Thank you for the comments and recognition of our work!**
>
> ###  Q1.1 “In adaptive attacks, EOT is the only attack considered. …… one of the minimum knowledge that should be provided to any adaptive attacker.”
>
>
> Thanks for this constructive comment.
>
> Actually, we have also considered that the magnitude $\nu$ of defense noise is known to the attacker, as shown in Line 207 of Sec. 4.2.1. In this case, according to our theoretical analysis that "the large ratio $\frac{\nu}{\mu}$ leads the effectiveness of RND" (see Line 200), the adaptive strategy that the attacker can adopt is to increase $\mu$. The effect of this strategy should be analyzed from three perspectives.
>
> **1)** for the fixed $\nu$, the smaller $\frac{\nu}{\mu}$ corresponds to the lower query complexity (see Line 192), i.e., increasing the attack performance.
>
> **2)** increasing $\mu$ may cause the less accurate estimation of gradient in Eq.(3) or search direction Eq.(5), leading to a significant decrease in attack performance.
>
> **3)** as demonstrated in Line 188-192 in Remark 1, given the gap $\epsilon$ between $f_{\mu, \nu}(\mathbf{x})$ and $f_{\nu}(\mathbf{x})$, $\mu$ should satisfy $\mu < \frac{\epsilon}{d^{1/2} L_0(f)}$. In other words, increasing $\mu$ will enlarge the gap $\epsilon$. Consequently, the converged value of $f_{\mu, \nu}(\mathbf{x})$ may be far from $f_{\nu}(\mathbf{x})$, leading to poor attack performance.
>
> This is also observed in our experiments. As shown in Fig. 3, when $\log_{10}(\mu) > -2$, the attack failure rates of NES, ZOsignSGD, and Bandit sharply increase. And the experimental results in Table 9 of the supplementary also show when the $\mu$ of Square attack is larger than 0.3, the attack performance will decrease. Thus, even the attacker knows $\nu$, he still cannot keep increasing $\mu$.
>
>
> Given fixed $\nu$, the only possible slight benefit for the attacker is that he can quickly identify a suitable range to adjust $\mu$ to achieve good attack performance. However, the defender can increase $\nu$ to shrink such a "suitable range", which can be implemented by the proposed RND-GF method (see Sec. 4.4).
>
> **Finally**, we should emphasize that in practice, the attacker is difficult to know or estimate the defense magnitude $\nu$. It is easy to know the defense strategy if observing the output difference between two same queries. However, due to the non-linearity of the model, it is difficult to estimate the input difference (or the noise magnitude) according to the output difference.
>
> ---
> ### Q1.2 “Moreover, all the theorems are heavily dependent …… when the knowledge is available to the attacker.”
> We are a little bit confused about the comment "Moreover, all the theorems .... when the knowledge is available to the attacker", such as the meaning of "how to do the results fair". We guess that one concern of the reviewer may be whether the defense performance is only related to the ratio $\frac{\nu}{\mu}$. As analyzed in Q1.1, the value of $\mu$ is also important. It would be greatly appreciated If the reviewer can provide further comments to help us to understand more correctly.
>
> ---
>
> ### Q2. Scope of work: "The scope of the work, though important ...... and is not applicable in the case of non-query attacks."
> Thanks for this insightful comment. Indeed our theoretical analysis only covers the defense to pure query-based attacks. We have provided some discussions about this limitation in Section 5.5 (see Line 366-372). We would like to further demonstrate that we have not found rigorous theoretical analysis about the defense against black-box attacks utilizing adversarial transferability between models, possibly due to the difficulty of accurately modeling the adversarial transferability. And, since the mechanisms of query-based and transfer-based attacks are significantly different, our current analysis about query-based cannot directly cover transfer-based attacks.
>
> However, as the first theoretical analysis of black-box defense, we believe that this work could attract broad interest in the research community and may inspire more researchers to further explore black-box defense.
>
> Besides, as discussed in Section 5.5, there have been some practical defense strategies [1][2] developed for transfer-based attacks. RND is very easy to combine with these methods to form a complementary defense system. It is interesting to explore the combination of RND and these works. Due to space limits, we leave it to future works.
>
>
> [1] Improving adversarial robustness via promoting ensemble diversity, Pang et al., ICML 2019
>
> [2] Dverge: Diversifying vulnerabilities for enhanced robust generation of ensembles, Yang et al., NeurIPS 2020
>
>
> ---
>
> ### Q3. Failure rates: "would you suggest that RND is still beneficial ...... clean accuracy as well)"
>
> We should first clarify the confusion in this comment. We guess that "at the optimal selection of $\mu$ ... the failure rates can be very high" should be "the failure rates can be very low", corresponding to the meaning that the optimal $\mu$ can lead to very low defense performance. Based on this understanding, our responses include two points, utilizing the result curves shown in Figs. 1,2,3.
>
> **1)**  the failure rates of RND are always higher than that of without RND, explicitly demonstrating the benefit of RND to model robustness.
>
> **2)**  comparing "RND $\nu$=0.02"(yellow curve) with "RND $\nu$=0.01" (blue curve), the failure rates with higher $\nu$ are always higher than that with lower $\nu$, which is exactly aligned with our theoretical analysis. It implies that the defender can add higher $\nu$ to increase the possible lowest failure rate, and the proposed RND-GF can further alleviate the harm of the higher $\nu$ on the clean accuracy.
>
> Thus, RND is always beneficial to model robustness, and the possible lowest failure rate could be increased by suitably adjusting $\nu$. For the analysis of the influence of $\mu$, please see our response to Q1.1.

---

> > ### Author Response · Authors · 2021-08-22
> > **Hope our posted response can help to address your concerns**
> >
> > Dear Reviewer RFP8,
> >
> > We sincerely hope our posted response can help to address the lingering points of concerns and your re-assessment of our work. If you have any further comment and question, please let us know and we are glad to write a follow-up response.
> >
> > Sincerely,
> >
> > Authors

---

> > ### Author Response · Authors · 2021-09-02
> > **We are glad to provide further responses to any remaining concern**
> >
> > Dear Reviewer RFP8,
> >
> > We sincerely hope our posted responses can help to address your concerns. Although the discussion phase is due soon, we are still very glad to provide further responses to any remaining concern. That will be greatly appreciated.
> >
> > Sincerely,
> >
> > Authors

---

### Decision · Program_Chairs · 2021-09-27

**Decision:**

Accept (Poster)

**Comment:**

This paper shows that adding a small amount of random noise to the output of a neural network can prevent current black-box attacks. The theory is convincing, the experiments cover a wide range of attacks, and the reviewers are satisfied by the author response. Even though there are still some questions about the generality of the proposed defense and if it could be evaded by stronger attacks, the proposal is interesting and will motivate future attack research to improve on randomized defenses.